# Spatiotemporal distributionally robust optimization for improved cross-patient EEG seizure analysis

## Abstract

Automatic seizure detection and classification from electroencephalography (EEG) hold significant potential to enhance epilepsy diagnosis and treatment. However, deep learning approaches often suffer from limited generalization ability to unseen patients due to inter-patient variability in EEG. While existing studies primarily focus on model architecture design or pre-training strategies to alleviate the problem, the optimization framework for robust cross-patient generalization, especially under the inherently spatiotemporal structure of EEG, remains underexplored. In this work, we propose SpatioTemporal Distributionally Robust Optimization (STDRO), a novel method to improve cross-patient seizure analysis in parallel to existing architectural/pre-training solutions. STDRO constructs and learns structured uncertainty sets that explicitly capture the spatial and temporal characteristics of EEG signals, thereby inducing data-adaptive worst-case distributions for robust optimization and improving cross-patient generalization. Extensive experiments demonstrate the effectiveness of STDRO as a plug-and-play approach to consistently enhance state-of-the-art seizure detection and classification models across diverse evaluation scenarios. Our work advances robust EEG-based seizure analysis toward practical applications with cross-patient scenarios.

## 1 Introduction

Automatic seizure analysis is a critical task for assisting and accelerating clinical diagnosis and effective treatment, as epilepsy seizure is a chronic neurologic disorder affecting nearly 50 million people worldwide (Mormann et al., 2007; World Health Organization, 2016). Currently, electroencephalography (EEG) remains a primary tool for identifying and characterizing seizures in clinical practice (Ahmad et al., 2016).

Recent progress in automatic EEG-based seizure analysis has been primarily driven by deep learning approaches (Tang et al., 2023; Afzal et al., 2024; Gui et al., 2024). However, a major challenge lies in the cross-patient setting, where the subjects in the training set differ from those in the test set (Zhou et al., 2022; Tang et al., 2022; Zhang et al., 2024a). This scenario closely reflects real-world clinical practice, where models should be able to generalize to new patients without retraining. The difficulty arises from substantial inter-patient variability in EEG patterns, along with the complex spatiotemporal structure of EEG signals. Achieving robust performance under these conditions requires models to maintain strong generalization across diverse patient distributions.

To tackle the cross-patient challenge, existing works have explored several directions. The prevailing paradigm follows a pretrain-finetune framework, e.g., DCRNN (Tang et al., 2022), VQMTM (Gui et al., 2024), and NeuroLM (Jiang et al., 2025). They leverage self-supervised pretraining on large data to encourage learning more generalizable representations, facilitating cross-patient generalization in downstream tasks. Another line of research focuses on innovative network architectures to enhance the performance or efficiency of EEG analysis (Peng et al., 2022; Tang et al., 2023; Afzal et al., 2024; Hong et al., 2025). However, these works do not consider the fundamental optimization process of models. Some works explore robust representation learning to improve cross-patient generalization, such as through adversarial learning (Zhang et al., 2020; 2024a) or invariant repre-

sentation learning (Wu et al., 2024). Nevertheless, they often rely on strong assumptions (Rosenfeld et al., 2021) and presuppose the existence of fully invariant representations across patients, which may be unrealistic, and risk over-invariance with degraded performance (Jiaqi et al., 2025).

Distributionally Robust Optimization (DRO) is a promising optimization tool to improve the generalizability of the model for unseen data (Kuhn et al., 2025; Chen et al., 2020; Sinha et al., 2018; Rahimian & Mehrotra, 2019). Unlike invariance-based approaches, DRO optimizes performance under the worst-case distribution within a defined uncertainty set that aims to cover possible distribution shifts. A key problem of DRO is how to properly specify this uncertainty set. Many existing methods manually instantiate it as a ball under a distributional distance metric (e.g., Wasserstein distance (Sinha et al., 2018; Mohajerin Esfahani & Kuhn, 2018) or maximum mean discrepancy (Staib & Jegelka, 2019)). While analytically convenient, such sets can be either overly conservative for generalization or insufficiently protective for feasible learning under the worst case in real applications (Frogner et al., 2021; Sagawa et al., 2020). This particularly brings challenges to EEG seizure analysis with complex spatiotemporal data structures.

Specifically, the key challenge for DRO lies in constructing a practical uncertainty set that properly adapts to the properties and structures of EEG data. Since EEG signals are composed of dynamic time series across spatial channels, with spatial autocorrelation across brain regions and temporal continuity (Gloor et al., 1990; Amor et al., 2009; Tang et al., 2023), the uncertainty set should reflect such structures to better capture potentially plausible distributional shifts adaptive to data, which is rarely studied in DRO. Apart from the spatiotemporal structure, the uncertainty set should also consider the stability to characterize the significant variability, where components exhibiting higher variance on the objective require larger perturbation ranges. Jointly accounting for the data structure and stability is critical for constructing a practically better uncertainty set.

In this work, we propose the SpatioTemporal Distributionally Robust Optimization (STDRO) algorithm to solve the cross-patient problem for EEG-based seizure analysis. The key is to build and learn the uncertainty set with both the spatiotemporal structure of EEG and a surrogate stability objective. First, we initially construct the uncertainty set for EEG data with spatial correlation graphs and temporarily evolving properties, incorporating data structure into distribution modeling. Second, we leverage a stability objective, mainly characterized by the gap between worst and best performance across patient groups, to optimize the uncertainty set. Third, we further enforce the spatiotemporal properties, i.e., the spatial connectivity and temporal continuity, during the optimization of the uncertainty set. In this way, STDRO synergizes the spatiotemporal structure and stability of EEG for data-adaptive distribution modeling, facilitating robust optimization for cross-patient generalization. Notably, our method is complementary to most existing methods such as pretrain-finetune or architecture design. Extensive experiments on various datasets and settings demonstrate the effectiveness of STDRO as a plug-and-play approach to further improve state-of-the-art approaches' performance in seizure detection and classification tasks, advancing robust EEG-based seizure analysis in cross-patient scenarios.

## 2 RELATED WORK

Automatic epileptic seizure analysis has been propelled by deep learning, with a large body of work advancing network design and representation learning. For the network architecture, graph-based models are popular for capturing the non-Euclidean structure of multichannel EEG (Chen et al., 2025; Klepl et al., 2024; Tang et al., 2022). For example, Dist-DCRNN (Tang et al., 2022) combines graph diffusion convolutional recurrent neural network with self-supervised pretraining, and Graphs4former (Tang et al., 2023) introduces the combination of graph neural networks and Structured State Space models to capture long-range spatiotemporal dependencies, achieving remarkable performance even for cross-patient settings. Other architectural directions include Tie-EEGNet with a temporal information enhancement module (Peng et al., 2022), ConvLSTM for spatiotemporal modeling (Yang et al., 2022), densely connected inception-style CNNs trained with weak labels (Saab et al., 2020), neural memory networks with plasticity (Ahmedt-Aristizabal et al., 2020), spiking neural networks tailored for efficiency (Shan et al., 2023; Zhang et al., 2024b), and KAleepNet based on Kolmogorov–Arnold networks (Akbar et al., 2025). Along with network architectures, self-supervised representation learning emerges as a powerful approach, with many works focusing on jointly designing network structure and pre-training, such as VQMTM (Gui et al., 2024) with vec-

tor quantization masked time-series modeling and BERT-style self-supervised learning for the EEG time series data analysis. NeuroLM (Jiang et al., 2025) further leverages the capabilities of pre-trained Large Language Models (LLMs) by regarding EEG signals as a foreign language to enhance the model's multi-task and inference capabilities. These advances principally target architecture or pretraining rather than the optimization approaches.

Besides architecture and pre-training, there are also other works to deal with the cross-patient problem. To mitigate inter-patient variability, domain generalization and adaptation techniques have been explored, such as invariant representation learning (Wu et al., 2024; Zhang et al., 2023), adversarial learning (Zhang et al., 2024a; Ayodele et al., 2020), feature disentanglement(Feng et al., 2026; Zhao et al., 2022; Zhang et al., 2020), as well as domain adaptation across datasets (Fan et al., 2024; Xia et al., 2022; Nasiri & Clifford, 2021; He & Wu, 2020). Meta-learning has also been explored to simulate distribution shifts across episodes and facilitate rapid adaptation (Liu et al., 2025; Zhu et al., 2020; Duan et al., 2020). As a parallel technique, some works also investigated data augmentation methods (Shu et al., 2024; Wang et al., 2023; Peng et al., 2022; Gómez et al., 2020; Wei et al., 2019). In EEG decoding, there are also works exploring cross-subject problems under the perspective of online continula learning (Duan et al., 2023a). Among current approaches, the self-supervised pre-training (Gui et al., 2024; Yuan et al., 2023; Tang et al., 2022) remains the state-of-the-art method in large-scale settings. There is limited work to consider robust optimization to improve the cross-patient generalization.

Distributionally robust optimization offers a principled optimization framework that targets worst-case performance over an uncertainty set designed to capture potential train-to-test shifts (Kuhn et al., 2025; Chen et al., 2020; Rahimian & Mehrotra, 2019; Sinha et al., 2018). Classical uncertainty sets are often specified via moment constraints (Delage & Ye, 2010; Bertsimas et al., 2018), $f$-divergences (Sagawa et al., 2020; Namkoong & Duchi, 2016), or Wasserstein balls (Sinha et al., 2018; Mohajerin Esfahani & Kuhn, 2018). While analytically convenient, such generic sets can be overly conservative or misaligned with real-world shifts, limiting practical gains (Frogner et al., 2021; Hu et al., 2018). Some works try to adapt uncertainty sets with data-driven approaches (Liu et al., 2021; 2022), but they do not delve into the specific structures of data, such as the inherent spatiotemporal structure of EEG signals. Duan et al. (2023b) explored DRO in EEG decoding tasks by introducing dynamically evolved data distributions via Wasserstein gradient flows, while their approach does not exploit the intrinsic spatiotemporal structure of EEG signals as our method. For EEG seizure analysis, little work has explored optimization-centric, structure-aware DRO in this domain.

## 3 METHOD

### 3.1 PROBLEM FORMULATION

We first introduce the problem formulation of cross-patient seizure detection and classification tasks.

**Seizure detection and classification** Given a period of multivariate EEG signal $\boldsymbol{X} \in \mathbb{R}^{C \times T}$ from a patient with $T$ time steps and $C$ spatial channels (electrodes), we aim to construct a machine learning model $f(\theta)$ to predict the seizure label $y$ for $\boldsymbol{X}$. The seizure detection task aims to automatically classify the seizure and non-seizure periods from the EEG of epilepsy patients, so $y \in \{0, 1\}$ and it is a binary classification problem. The seizure classification task aims to classify the seizure type of the seizure EEG periods, therefore, $y$ is the multiclass label in the seizure classification task.

**Cross-patient setting** We aim to build the robust model under the cross-patient setting, where the patients $N_{train}$ in the training dataset $P_0$ are totally different from the testing patients $N_{test}$, i.e. $N_{train} \cap N_{test} = \emptyset$. The key difficulty of cross-patient seizure detection is the model's generalization ability challege introduced by the variation of different individuals. Following a prior study (Tang et al., 2022), we examine our model's capability for fast detection and classification over EEG clips with different time window sizes.

### 3.1.1 FORMULATION AS DISTRIBUTIONALLY ROBUST OPTIMIZATION

Under the cross-patient setting, the model requires generalization to new populations with distribution shift, so we formulate the problem as a distributionally robust optimization problem and optimize the model for the worst-case performance. Specifically, the target generalized populations

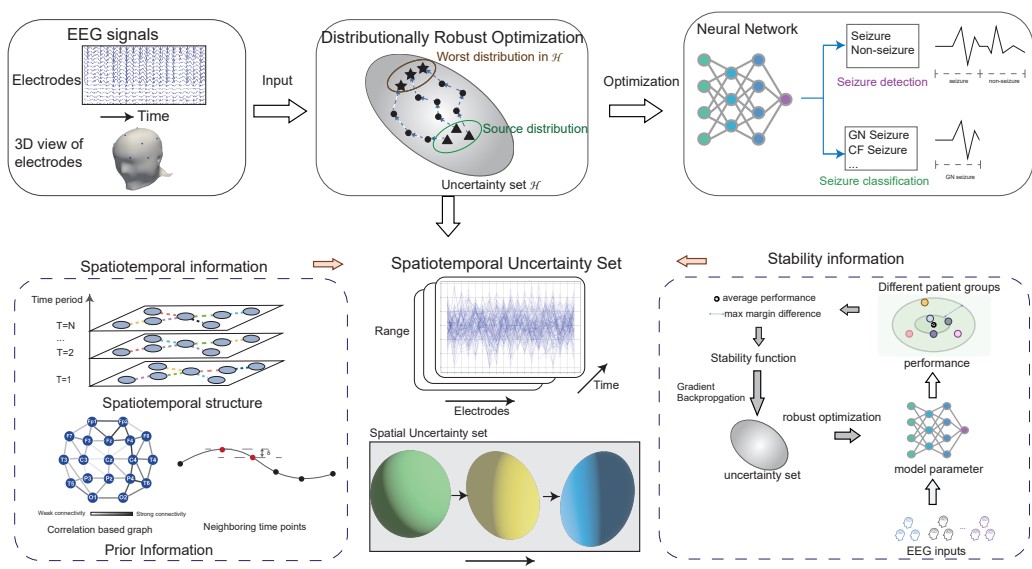

Figure 1: Overview of the proposed spatiotemporal distributionally robust optimization. DRO optimizes neural networks under the worst-case distribution within the defined uncertainty set. STDRO constructs and learns the critical uncertainty set by incorporating the spatiotemporal characteristics of EEG signals and the stability objective.

are represented as the uncertainty set $\mathcal{H} = \{Q|\text{dist}(Q, P_0) < \rho\}$, where $\rho$ is a hyperparameter indicating the distribution shift strength and $\text{dist}$ is a function to characterize the distribution distance. Then the optimization problem of our classification model can be formulated as:

$$\min_{\theta \in \Theta} \sup_{Q:\text{dist}(Q,P_0)\leq\rho} \mathbb{E}_{\boldsymbol{X},Y\sim Q}[\ell(\theta; \boldsymbol{X}, Y)],$$

where $P_0$ is the training distribution, $\theta$ is the model parameter, $\boldsymbol{X}$ is the EEG signals and $Y$ is the seizure label. The Wasserstein distance function $\text{dist}(Q, P_0) = W_c(Q, P_0) = \inf_{M \in \Pi(Q,P_0)} \mathbb{E}_{(z,z')\sim M}[c(z, z')]$, where $c : \mathcal{X} \times \S \to [0, \infty)$ is the transportation cost function. $\ell$ is the cross-entropy loss function for binary or multi-class classification. DRO leverages adversarial perturbation to get the worst-case distribution, i.e., solving the inner optimization problem to obtain data points for model optimization.

## 3.2 SPATIOTEMPORAL DISTRIBUTIONALLY ROBUST OPTIMIZATION (STDRO)

Figure 1 shows the schematic diagram of our proposed spatiotemporal distributionally robust optimization (STDRO) method. The core idea of our approach is to construct a more practical spatiotemporal uncertainty set for EEG signals by incorporating the structure and stability information of the data. Specifically, the uncertainty set will dynamically change over time while containing spatial correlation information and, meanwhile, will be optimized based on the stability information under the spatio-temporal constraints. We will elaborate on each part in Sections 3.2.1 and 3.2.2.

### 3.2.1 SPATIOTEMPORALLY STRUCTURED UNCERTAINTY SET

EEG signals have spatial and temporal dimensions. Specifically, epileptic brain activity is not confined to a single brain region, instead, it involves different brain regions that are spatially distributed and have functional connectivity (Tang et al., 2023). The EEG signals of different electrodes and their underlying connectivity are changing dynamically over time (Gloor et al., 1990; Amor et al., 2009).

To better capture the dynamic structure into the uncertainty set, we first split the EEG signals into $N$ short time periods, and each time period has $L = \frac{T}{N}$ time steps. During each period $t$, we character-

ize the uncertainty set through the current non-Euclidean spatial structure among different EEG electrodes, and we represent the uncertainty set through the correlation graph $W_t$, where $W_t^{jk}$ is the absolute value of the normalized cross-correlation between the preprocessed signals in electrode $v_j$ and electrode $v_k$ among all EEG siganls at the $t$-th period, i.e. $W_t^{jk} = |X_{j,\frac{(t-1)*T}{N}:\frac{t*T}{N}} * X_{k,\frac{(t-1)*T}{N}:\frac{t*T}{N}}|$, $1 \le t \le N, 1 \le j \le C, 1 \le k \le C$. $W_t$ controls the shape of the uncertainty set in $t$-th period.

Then the initial spatiotemporal uncertainty set can be represented by the dynamic graphs $\{W_t\}_{t=1}^{N}$, where the $t$-th time period of EEG signals' perturbation level are controlled by $W_t$. Thus the covariate weight $W \in \mathbb{R}^{(C \times T+1) \times (C \times T+1)}$ of the initial spatiotemporal uncertainty set can be formulated as

$$W = \begin{bmatrix} I_{\frac{T}{N}} \bigotimes \mathrm{diag}(W_1, W_2, \cdots, W_N) & \mathbf{0}_{(C*T) \times 1} \\ \mathbf{0}_{1 \times (C*T)} & 1 \end{bmatrix},$$

where $\bigotimes$ represents the element-wise tensor product. Based on it, the intial spatiotemporal uncertainty set can be formulated as $H = \{Q : W_{c_w}(Q, P_0) \le \rho\}$, where $W_{c_w}$ denotes the Wasserstein distance with the transportation cost function $c_w$ defined as

$$c_w(z, z') = (z - z')^T W (z - z').  \tag{1}$$

We further provide a theoretical analysis of the proposed spatiotemporal uncertainty set in Appendix F to show that when the spatiotemporal structure captures the intrinsic low-dimensional manifold of EEG data, the distributionally robust generalization bound can be tightened compared to standard DRO with Wasserstein balls. Complete details are given in Appendix F.

During training, the spatiotemporal uncertainty set will be learnable so that it can incorporate stability information as will be introduced below. To ensure that the ever-changing uncertainty set still contains the spatiotemporal information, we restrict the shape of the covariate weight in the uncertainty set as

$$W = \begin{bmatrix} I_{\frac{T}{N}} \bigotimes \mathrm{diag}(W_1 M_1, W_2 M_2, \cdots, W_N M_N) & \mathbf{0}_{(C \times T) \times 1} \\ \mathbf{0}_{1 \times (C \times T)} & 1 \end{bmatrix}$$

where $W_i$ is fixed while $M_i$ is learnable and initialzed as $I$. Then the spatial information embedded in the covariate weight can ensure a similar distribution of correlated channels in the uncertainty set. In addition, to incorporate the prior that neighboring temporal distributions are similar, we design a loss to constrain the weights of neighboring times $\mathcal{H} = \|W_t M_t - W_{t+1} M_{t+1}\|_F$ to be close, as will be introduced below.

### 3.2.2 STABILITY-INDUCED UNCERTAINTY SET

As mentioned above, the constructed uncertainty set in our method not only has a dynamic spatiotemporal structure but also incorporates stability information that can be learned from data automatically. We will introduce the learning process of the EEG uncertainty set in this section.

In the uncertainty set of our method, the transportation cost function can be formulated as

$$c_w(z, z') = (X - X')^T I_{\frac{T}{N}} \bigotimes \mathrm{diag}(W_1 M_1, W_2 M_2, \cdots, W_N M_N)(X - X') + \infty \times \mathbb{I}_{y \ne y'} \tag{2}$$

$$= \sum_{t=1}^{N} (X_{:,(t-1) \times L:t \times L} - X'_{:,(t-1) \times L:t \times L})^T W_t M_t (X_{:,(t-1) \times L:t \times L} - X'_{:,(t-1) \times L:t \times L}) + \infty \times \mathbb{I}_{y \ne y'},$$

where the covariate weight $A_t = W_t M_t$ controls the perturbation level of each dimension in the EEG uncertainty set of the $t$-th period. The higher the weight element value is, the lower perturbation will be imposed in this dimension.

Ideally, we hope the range of different data dimensions (i.e. channels multiplied by time) in EEG uncertainty set is heterogeneous, and the dimension with higher "variance" has a larger range in the uncertainty set for EEG signals. To better capture the dimensional "variance", we leverage the idea of feature stability across environments as a surrogate objective. We first group the training patients

into different environments (patient groups) $\mathcal{E}_{tr}$ according to the EEG signals. Specifically, we average the EEG signals of each single patient, and conduct clustering algorithms on these averaged EEG signals. Then, since the features that minimally affect the performance difference among environments are stable with low variability, the maximum margin among different patient groups $\max_{e_p, e_q \in \mathcal{E}_{tr}}(\ell^{e_p}(\theta(\boldsymbol{W})) - \ell^{e_q}(\theta(\boldsymbol{W})))$ can reflect the stability, where the model parameter $\theta$ is the function of uncertainty set's covariate weight $\boldsymbol{W}$ under the DRO framework. So we design the loss function $\max_{e_p, e_q \in \mathcal{E}_{tr}}(\ell^{e_p}(\theta(\boldsymbol{W})) - \ell^{e_q}(\theta(\boldsymbol{W})))$ to adapt $\boldsymbol{W}$. Additionally, we expect the model learned from the constructed uncertainty set has good performance across all patient groups, so we come up with another objective function $\frac{1}{|\mathcal{E}_{tr}|}\sum_{e \in \mathcal{E}_{tr}} \ell^e(\theta(\boldsymbol{W}))$ to adapt $\boldsymbol{W}$.

Overall, the objective function of our STDRO method can be summarized as:

$$\min_{\theta \in \Theta} \sup_{Q: W_{c_w}(Q, P_0) \leq \rho} \mathbb{E}_{\boldsymbol{X}, Y \sim Q}[\ell(\theta; X, Y)],$$

$$\text{s.t. } \boldsymbol{M} \in \arg \min_{\boldsymbol{M} \in \mathcal{M}} \frac{\sum_{e \in \mathcal{E}_{tr}} \ell^e(\theta)}{|\mathcal{E}_{tr}|} + \alpha \max_{e_p, e_q \in \mathcal{E}_{tr}}(\ell^{e_p} - \ell^{e_q}) + \beta \sum_t \|\boldsymbol{W}_{t+1}\boldsymbol{M}_{t+1} - \boldsymbol{W}_t\boldsymbol{M}_t\|_F^2 \tag{3}$$

where $W_{c_w}$ denotes the Wasserstein distance with transportation cost function $c_w$ defined as

$$c_w(z, z') = (z - z')^T \begin{bmatrix} \boldsymbol{I}_{\frac{T}{N}} \bigotimes \text{diag}(\boldsymbol{W}_1\boldsymbol{M}_1, \boldsymbol{W}_2\boldsymbol{M}_2, \cdots, \boldsymbol{W}_N\boldsymbol{M}_N) & \boldsymbol{0}_{(C \times T) \times 1} \\ \boldsymbol{0}_{1 \times (C \times T)} & 1 \end{bmatrix} (z - z').$$

$$\mathcal{M} = \{\{\boldsymbol{M}_1, \boldsymbol{M}_2, .., \boldsymbol{M}_N\} : \text{diag}(\boldsymbol{W}_t\boldsymbol{M}_t) \succeq 0, 1 \leq t \leq N\}.$$

$\|\boldsymbol{W}_{t+1}\boldsymbol{M}_{t+1} - \boldsymbol{W}_t\boldsymbol{M}_t\|_F^2$ denotes the similar distribution restriction between adjacent time periods in EEG signals, where $\|\cdot\|_F$ denotes the Frobenius norm. $\frac{\sum_{e \in \mathcal{E}_{tr}} \ell^e(\theta)}{|\mathcal{E}_{tr}|}$ are the average loss across environments $\mathcal{E}_{tr}$. $\max_{e_p, e_q \in \mathcal{E}_{tr}}(\ell^{e_p} - \ell^{e_q})$ measures the feature stability. $\alpha$ and $\beta$ are the hyperparameters that adjust the tradeoff among average performance, feature stability, and time continuity. The architecture of the model parameterized by $\theta$ is arbitrary and can be any type of neural network. During the learning process, $\boldsymbol{W}_t$ is fixed while $\boldsymbol{M}_t$ is learnable. $\boldsymbol{M}$ refers to any $\boldsymbol{M}_t, 1 \leq t \leq N$.

### 3.3 OVERALL OPTIMIZATION PROCEDURE

The whole objective function is a bi-level optimization problem, and the optimization of model parameter $\theta$ and weight $\boldsymbol{M}$ is performed alternately. Given the current $\boldsymbol{M}$, the objective function of model parameter $\theta$ is $\min_{\theta \in \Theta} \sup_{Q: W_{c_w}(Q, P_0) \leq \rho} \mathbb{E}_{\boldsymbol{X}, Y \sim Q}[\ell(\theta; X, Y)]$, which can be reformulated as below through the Lagrangian relaxation (Sinha et al., 2018):

$$\min_{\theta \in \Theta} \sup_Q \{\mathbb{E}_{X \sim Q}[\ell(\theta; \boldsymbol{X}, Y)] - \lambda W_{c_w}(Q, P_0)\}. \tag{4}$$

For simplicity in notation, we denote $S_\lambda(\theta; (\boldsymbol{X}, y)) = \sup_{\varepsilon \in Z}(l(\theta, \varepsilon) - \lambda c_w(\varepsilon, (\boldsymbol{X}, y)))$. This problem can be solved through adversarial optimization, and we denote the approximation maximizer solution of $S_\lambda(\theta, \boldsymbol{X}, Y)$ as $\widetilde{\boldsymbol{X}}$. Then we can optimize the model parameter $\theta$ using $(\widetilde{\boldsymbol{X}}, Y)$.

During the optimization of weight $\boldsymbol{M}$, the objective function is a multi-environment objective $R(\theta)$.

$$R(\theta) = \frac{\sum_{e \in \mathcal{E}_{tr}} \ell^e(\theta)}{|\mathcal{E}_{tr}|} + \alpha \max_{e_p, e_q \in \mathcal{E}_{tr}}(\ell^{e_p} - \ell^{e_q}) + \beta \sum_{t=1}^{N-1} \|\boldsymbol{W}_t\boldsymbol{M}_t - \boldsymbol{W}_{t+1}\boldsymbol{M}_{t+1}\|_F. \tag{5}$$

We denote $A(\theta) = \frac{\sum_{e \in \mathcal{E}_{tr}} \ell^e(\theta)}{|\mathcal{E}_{tr}|} + \alpha \max_{e_p, e_q \in \mathcal{E}_{tr}}(\ell^{e_p} - \ell^{e_q})$. The weight $\boldsymbol{M}$ is updated through gradient descent, and $\frac{\partial R(\theta(\boldsymbol{W}(\boldsymbol{M})))}{\partial \boldsymbol{M}}$ is approximated as following:

$$\frac{\partial R(\theta(\boldsymbol{W}(\boldsymbol{M})))}{\partial \boldsymbol{M}} = \frac{\partial A}{\partial \theta}\frac{\partial \theta}{\partial \widetilde{\boldsymbol{X}}}\frac{\partial \widetilde{\boldsymbol{X}}}{\partial \boldsymbol{W}}\frac{\partial \boldsymbol{W}}{\partial \boldsymbol{M}} + \beta\frac{\partial \sum_{t=1}^{N-1} \|\boldsymbol{W}_t\boldsymbol{M}_t - \boldsymbol{W}_{t+1}\boldsymbol{M}_{t+1}\|_F}{\partial \boldsymbol{M}}. \tag{6}$$

Among these components, $\frac{\partial A}{\partial \theta}$ and $\frac{\partial \boldsymbol{W}}{\partial \boldsymbol{M}}$ of the first term and the second term can be calculated easily. $\frac{\partial \theta}{\partial \widetilde{\boldsymbol{X}}}$ of the first term can be approximated through the gradient descent of $\theta$ as

$$\frac{\partial \theta}{\partial \widetilde{\boldsymbol{X}}} \approx -\epsilon_\theta \sum_i \frac{\nabla_\theta \hat{\ell}(\theta^i; \widetilde{\boldsymbol{X}}, Y)}{\partial \widetilde{\boldsymbol{X}}}, \tag{7}$$

Table 1: Comparisons results of different methods for 12s EEG-based cross-patient seizure detection on the TUSZ dataset.

| Method | AUROC (%) | F1-score (%) | Accuracy (%) | Recall (%) | Precision (%) |
|---|---|---|---|---|---|
| TieEEG Net | 68.7 | 30.3 | 73.5 | 52.8 | 21.3 |
| CNN-LSTM | 70.5 | 29.3 | 75.7 | 46.2 | 21.5 |
| LSTM | 77.6 | 36.5 | 81.3 | 49.2 | 29.0 |
| Dense-CNN | 78.0 | 32.6 | 85.8 | 40.1 | 36.6 |
| VQ-MTM | 79.2 | 42.0 | 88.5 | 41.0 | 43.1 |
| VQ-MTM+STDRO (ours) | **80.0** | 40.6 | 90.6 | 31.9 | 55.9 |
| GraphS4former | 85.7 | 50.5 | 85.8 | 66.1 | 40.8 |
| GraphS4former+STDRO (ours) | **87.1** | **52.2** | 87.4 | 63.0 | 44.5 |
| DCRNN | 86.7 | 50.8 | 87.8 | 57.6 | 45.4 |
| DCRNN+STDRO (ours) | **88.2** | **54.8** | 89.5 | 58.1 | 51.8 |

where $i$ is the iteration index, $\epsilon_\theta$ is the learning rate of $\theta$. And $\frac{\nabla_\theta \hat{\ell}(\theta^i; \widetilde{X}, Y)}{\partial \widetilde{X}}$ can be calculated through training. The third term $\frac{\partial \widetilde{X}}{\partial W}$ can be approximated during the adversarial process as

$$\frac{\partial \widetilde{X}}{\partial W} \approx -2\epsilon_x \lambda \sum_i \mathrm{Diag}(\widetilde{X}^i - X). \tag{8}$$

The details can be found in the Appendix, and the whole optimization process is illustrated in Algorithm 1.

## 4 EXPERIMENT

### 4.1 DATASET AND PREPROCESSING

We use several publicly available datasets: the Temple University Hospital EEG Seizure Corpus (TUSZ) V1.5.2 dataset (Obeid & Picone, 2016; Shah et al., 2018) and the CHB-MIT dataset (Goldberger et al., 2000) for seizure detection and classification. Meanwhile, we also evaluate the extension to the Dreem Open Dataset-Healthy (DOD-H) dataset (Guillot et al., 2020) for EEG-based sleep stage classification.

TUSZ dataset is the largest EEG database, which includes more than 5000 EEG files recorded by 19 electrodes in the traditional 10-20 systems. The dataset has both detection and seizure classification labels. It contains more than 900 hours of seizure duration. And there are four seizure classes in total: combined focal (CF), generalized non-specific (GN), absence (AB), and CT seizures. CHB-MIT dataset was collected by the Children's Hospital Boston focusing on the annotation of seizure detection. It is recorded by the traditional 10-20 systems. For our study, we analysed 21-channel EEGs of 23 patients in CHB-MIT. It contains more than 3 hours of seizure duration. DOD-H dataset has 16 polysomnographic (PSG) sensors. The sampling rate is 250 Hz. Each 30s-signal has 7500 time steps. And there are five sleep stages: wake, rapid eye movement (REM), non-REM sleep stages, N1, N2, and N3. The data preprocessing details can be found in the Appendix.

### 4.2 BASELINES AND EVALUATION METRICS

The baseline methods mainly include (1) pre-training finetune approaches: DCRNN (Tang et al., 2022) and VQ-MTM (Gui et al., 2024); and (2) architecture-based methods: Dense-CNN (Saab et al., 2020), LSTM (Graves, 2012), CNN-LSTM (Ahmedt-Aristizabal et al., 2020), TieEEG Net (Peng et al., 2022), and graphs4former (Tang et al., 2023). Additionally, we also evaluate and compare with the adversarial learning method PANN (Zhang et al., 2024a) for invariant feature learning and further show that our method is complementary to it as well. For all these baseline methods, we utilize their officially released code and adopt the suggested training strategies and hyperparameter settings in their original papers.

We first evaluate the baselines on the datasets and then apply our method to the three approaches with the highest performance. For the methods that follow the pretrain-finetune paradigm (Tang

Table 2: Comparisons results of different methods for 60s EEG-based cross-patient seizure detection on the TUSZ dataset.

| Method | AUROC (%) | F1-score (%) | Accuracy (%) | Recall (%) | Precision (%) |
|---|---|---|---|---|---|
| TieEEG Net | 59.4 | 28.8 | 62.6 | 51.5 | 20.0 |
| CNN-LSTM | 62.5 | 28.0 | 71.8 | 37.4 | 22.4 |
| LSTM | 70.8 | 35.8 | 69.8 | 57.2 | 26.0 |
| Dense-CNN | 81.7 | 50.5 | 84.8 | 52.9 | 48.3 |
| VQ-MTM | 81.8 | 51.7 | 87.6 | 46.5 | 58.2 |
| VQ-MTM+STDRO (ours) | **81.9** | **52.1** | 87.8 | 46.5 | 59.3 |
| DCRNN | 87.8 | 56.4 | 84.4 | 68.6 | 47.9 |
| DCRNN+STDRO (ours) | **88.3** | **61.7** | 88.1 | 65.6 | 58.3 |
| GraphS4former | 89.5 | 58.2 | 91.6 | 82.1 | 45.1 |
| GraphS4former+STDRO (ours) | **90.1** | **69.7** | 91.6 | 66.1 | 73.8 |

Table 3: Results of EEG-based cross-patient seizure classification on TUSZ dataset under 12s and 60s settings.

| Method | 12s | | 60s | |
|---|---|---|---|---|
| | F1-score (%) | Accuracy (%) | F1-score (%) | Accuracy (%) |
| GraphS4former | 53.4 | 64.3 | 63.8 | 69.1 |
| TieEEG Net | 56.1 | 65.1 | 63.7 | 67.2 |
| LSTM | 65.1 | 71.6 | 66.2 | 71.2 |
| Dense-CNN | 66.6 | 72.6 | 60.0 | 68.9 |
| Dense-CNN + STDRO (Ours) | **68.2** | **73.1** | **61.5** | **70.2** |
| DCRNN | 67.4 | 74.7 | 66.8 | 68.9 |
| DCRNN+STDRO (Ours) | **70.8** | **75.4** | **68.8** | **72.2** |
| VQ-MTM | 69.4 | 73.0 | 55.3 | 65.3 |
| VQ-MTM+STDRO (Ours) | **70.2** | **74.1** | **65.5** | **71.0** |

et al., 2022; Gui et al., 2024), we combine them with our method during the fine-tuning of models. For other models, we combine them with our method during the training of models. The covariate weight in our method is trained using the Adam optimizer (Kingma, 2015).

Following the common practice, we adopt weighted F1-score as the main evaluation metrics for seizure classification while we also report accuracy, and we leverage AUROC and F1-score as the main evaluation metrics for seizure detection while we also report accuracy, recall and precision.

### 4.3 MAIN RESULTS

The seizure detection results of different methods on the TUSZ dataset with 12s-EEG clips and 60s-EEG clips are presented in Table 1 and Table 2, respectively, and the seizure classification results on the TUSZ dataset is shown in Table 3. Table 4 shows the seizure detection results on the CHB-MIT dataset with 4s-EEG clips. Further, Table 5 shows the extended results to the sleep stage classification on the DOD-H dataset.

The results show that among various settings, tasks, and datasets, STDRO consistently boosts the performance of the state-of-the-art methods, demonstrating the effectiveness of our STDRO method. Particularly, STDRO significantly enhances the F1-score for the cross-patient seizure detection, and is complementary not only to pre-training/architecture approaches but also to other optimized-based methods such as adversarial learning for invariance. Meanwhile, STDRO can be successfully extended to other EEG-based BCI tasks. This validates the advantages of STDRO with incorporated spatiotemporal characteristics and stability of EEG signals for improved cross-patient generalization.

Table 4: Results of 4s EEG-based cross-patient seizure detection on CHB-MIT dataset.

| Method | AUROC (%) | F1-score (%) | Accuracy (%) | Recall (%) | Precision (%) |
|---|---|---|---|---|---|
| CNN-LSTM | 90.9 | 55.2 | 87.5 | 68.9 | 46.1 |
| DenseCNN | 93.5 | 65.4 | 90.6 | 79.6 | 55.4 |
| DCRNN | 94.0 | 69.3 | 93.7 | 63.9 | 75.7 |
| DCRNN+STDRO (ours) | **95.4** | **69.5** | 93.2 | 69.9 | 69.2 |
| DCRNN+PANN | 95.4 | 71.6 | 94.2 | 66.1 | 78.1 |
| DCRNN+PANN+STDRO (ours) | **96.2** | **71.9** | 93.9 | 70.0 | 73.8 |

Table 5: Results of cross-subject sleeping stage classification on the DOD-H dataset.

| Method | Macro-F1 | Kappa |
|---|---|---|
| LSTM | $0.609 \pm 0.034$ | $0.539 \pm 0.046$ |
| SimpleSleepNet | $0.720 \pm 0.001$ | $0.703 \pm 0.013$ |
| RobustSleepNet | $0.777 \pm 0.007$ | $0.758 \pm 0.008$ |
| DeepSleepNet | $0.716 \pm 0.025$ | $0.711 \pm 0.032$ |
| GraphS4former | $0.810 \pm 0.015$ | $0.790 \pm 0.020$ |
| GraphS4former+STDRO (ours) | $\mathbf{0.822 \pm 0.011}$ | $\mathbf{0.807 \pm 0.010}$ |

Table 6: Ablation study on 12s cross-patient seizure detection on the TUSZ dataset. Results are based on three runs of experiments.

| Method | AUROC | F1-score | Accuracy | Recall | Precision |
|---|---|---|---|---|---|
| DCRNN | $0.865 \pm 0.010$ | $0.495 \pm 0.013$ | $0.868 \pm 0.013$ | $0.589 \pm 0.030$ | $0.429 \pm 0.031$ |
| +DRO | $0.868 \pm 0.006$ | $0.514 \pm 0.001$ | $0.883 \pm 0.003$ | $0.564 \pm 0.009$ | $0.472 \pm 0.010$ |
| +DRO w/ spatiotemporal W | $0.874 \pm 0.005$ | $0.524 \pm 0.007$ | $0.878 \pm 0.001$ | $0.612 \pm 0.050$ | $0.462 \pm 0.036$ |
| +DRO w/ stability-induced W | $0.875 \pm 0.013$ | $0.521 \pm 0.034$ | $0.881 \pm 0.026$ | $0.585 \pm 0.077$ | $0.486 \pm 0.101$ |
| +STDRO (ours) | $0.876 \pm 0.008$ | $0.542 \pm 0.007$ | $0.893 \pm 0.002$ | $0.570 \pm 0.008$ | $0.510 \pm 0.008$ |

## 4.4 ABLATION STUDY

To verify the effect of each part of our method, we conduct an ablation study on the TUSZ Dataset. We choose the DCRNN as the model baseline, and compare several variants as follows: (1) DCRNN+DRO: The model is optimized by the vanilla distributionally robust learning where the uncertainty set is characterized by the Wasserstein distance; (2) DCRNN+DRO w/ spatiotemporal W: The uncertainty set of DRO is only constructed by the spatiotemporal structure; (3) DCRNN+DRO w/ stability-induced W: The uncertainty set is initialized by the Wasserstein distance and only optimized by the stability objective.

Results in Table 6 show the effectiveness of DRO and each component in our STDRO, validating the necessity of both the spatiotemporal structure of EEG data and the stability information. We also analyze the influence of the environment number in the stability objective in Table 7 and other hyerparameters $N, \alpha, \beta$ in Table 8, 9, 10, showing the robustness of our method.

## 4.5 VISUALIZATION OF COVARIATE WEIGHTS OF THE UNCERTAINTY SET

To illustrate the uncertainty set in our method, we visualize the covariate weight, which is utilized to characterize the high-dimensional uncertainty set (Section 3.2.1), for the 60s seizure classification task with DCRNN. As we split each EEG segment into four time periods (i.e., $N = 4$ in Section 3.2.1), we present the weight matrix for the first period and the difference from it for the remaining three periods for better visualization. As shown in Figure 2 (A-D), the weights have incorporated certain spatiotemporal structure in the data. Further, to characterize the stability-induced learning, we visualize the difference in the covariate weight of each time period before and after training in Figure 2 (E-H), demonstrating how the uncertainty set is influence by the stability objective.

## 4.6 ANALYSIS OF GROUPS AND CONFUSION MATRIX

We visualized the seizure clinical characteristics of different groups. As shown in Figure 3 (A), the groups formed by EEG clustering has different seizure-type proportion, indicating that the grouping

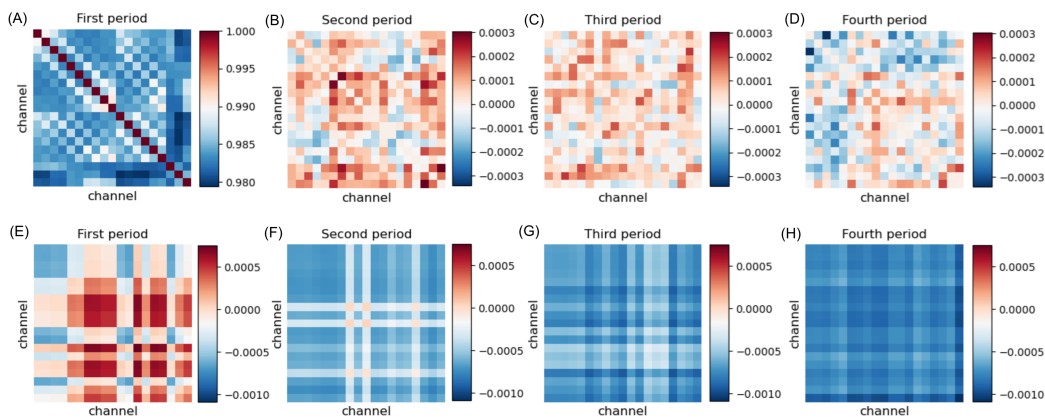

Figure 2: Learnable spatiotemporal covariate weight visualization. (A-D) The initial covariate weight and the difference between time periods. (E-H) The spatiotemporal covariate weight difference between initialization and training.

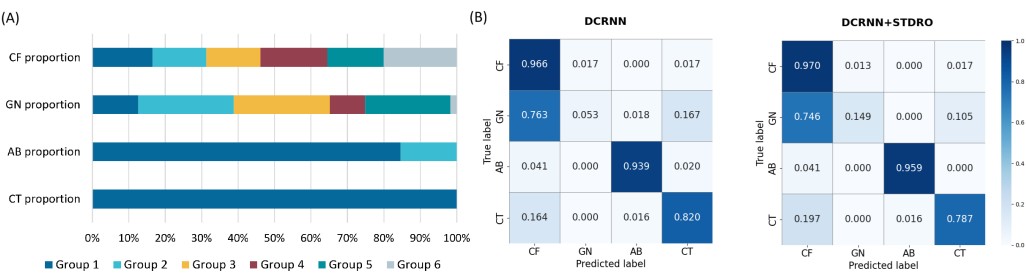

Figure 3: (A) The seizure clinical characteristics difference among clustered patient groups. (B) Confusion matrices for the DCRNN baseline without and with STDRO for 12-s seizure classification.

partially capture some features. In addition, We provided the confusion matrices for DCRNN baseline without and with STDRO for 12-s seizure classification in Figure 3 (B). As shown in the Figure, STDRO can improve the accuracy of the worst class (GN, from 0.053 to 0.149), whose improvement is larger than those of other classes.

## 5 CONCLUSION

In this work, we propose spatiotemporal distributionally robust optimization for cross-patient EEG-based seizure analysis. We introduce robust optimization as a complementary approach to existing pretrain-finetune or architecture design methods for cross-patient generalization. The proposed STDRO tackles the challenges of DRO for the inherent structure of EEG data by incorporating spatiotemporal structures and stability-induced information into the critical uncertainty set. Extensive experiments demonstrate the effectiveness of STDRO to further improve state-of-the-art methods across various settings in both seizure detection and classification, and validate the effectiveness of each component. Our work can advance robust seizure analysis toward practical cross-patient scenarios, and hold the potential for future extension to other time series data.

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

## A  MORE INTRODUCTION TO DRO AND SOLVING DRO

The objective function of a DRO problem is shown as the following:

$$\min_{\theta \in \Theta} \sup_{Q:\text{dist}(Q,P_0) \leq \rho} \mathbb{E}_{\boldsymbol{X},Y \sim Q}[\ell(\theta; \boldsymbol{X}, Y)].$$

In DRO, the uncertainty set $\{Q : \text{dist}(Q, P_0) \leq \rho\}$ defines a neighborhood around the input with perturbations. The objective is to optimize for the worst-case scenario within this neighborhood, ensuring good performance not only at the training distribution but also across neighborhood regions with shifts.

The inner optimization problem can be relaxed as

$$\sup_{Q} \{\mathbb{E}_{X \sim Q}[\ell(\theta; \boldsymbol{X}, Y)] - \lambda W_{c_w}(Q, P_0)\}$$

according to Largrangian relaxation. Given any data $(\boldsymbol{X}, y)$ in the training set $P_0$, the data samples of the worst distribution in the uncertainty set $(\widetilde{\boldsymbol{X}}, y)$ can be found through adversarial training (Sinha et al., 2018). Specifically, let $\widetilde{\boldsymbol{X}}$ denote the solution to maximizing $\ell(\theta; \widetilde{\boldsymbol{X}}, Y) - \lambda c_w(\widetilde{\boldsymbol{X}}, \boldsymbol{X})$, we adopt gradient descent to approximate $\widetilde{\boldsymbol{X}}$ in $m$ iterations, i.e.

$$\widetilde{\boldsymbol{X}}_{i+1} = \widetilde{\boldsymbol{X}}_i + \nabla_{\widetilde{\boldsymbol{X}}}(\ell(\theta; \widetilde{\boldsymbol{X}}, Y) - \lambda c_w(\widetilde{\boldsymbol{X}}, \boldsymbol{X})), 1 \leq i, i+1 \leq m.$$

In experiments, we set $m \in \{5, 10, 15, 20\}$.

## B  CALCULATION OF $\frac{\partial R(\theta(W(M)))}{\partial M}$

$\frac{\partial R(\theta(\boldsymbol{W}(\boldsymbol{M})))}{\partial \boldsymbol{M}}$ can be approximated through chain rule as the following:

$$\frac{\partial R(\theta(\boldsymbol{W}(\boldsymbol{M})))}{\partial \boldsymbol{M}} = \frac{\partial A}{\partial \theta}\frac{\partial \theta}{\partial \widetilde{\boldsymbol{X}}}\frac{\partial \widetilde{\boldsymbol{X}}}{\partial \boldsymbol{W}}\frac{\partial \boldsymbol{W}}{\partial \boldsymbol{M}} + \beta \frac{\partial \sum_{t=1}^{N-1} \|\boldsymbol{W}_t \boldsymbol{M}_t - \boldsymbol{W}_{t+1}\boldsymbol{M}_{t+1}\|_F}{\partial \boldsymbol{M}}. \quad (9)$$

Among these components, $\frac{\partial \theta}{\partial \widetilde{\boldsymbol{X}}}$ of the first term can be approximated through the gradient descent of $\theta$:

$$\theta^{i+1} = \theta^i - \epsilon_\theta \nabla_\theta \hat{\ell}(\theta^i; \widetilde{\boldsymbol{X}}, Y),$$

$$\frac{\partial \theta^{i+1}}{\partial \widetilde{\boldsymbol{X}}} = \frac{\partial \theta^i}{\partial \widetilde{\boldsymbol{X}}} - \epsilon_\theta \frac{\nabla_\theta \hat{\ell}(\theta^i; \widetilde{\boldsymbol{X}}, Y)}{\partial \widetilde{\boldsymbol{X}}},$$

$$\frac{\partial \theta}{\partial \widetilde{X}} \approx -\epsilon_\theta \sum_i \frac{\nabla_\theta \hat{\ell}(\theta^i; \widetilde{\boldsymbol{X}}, Y)}{\partial \widetilde{\boldsymbol{X}}}, \quad (10)$$

where $i$ is the iteration index, $\epsilon_\theta$ is the learning rate of $\theta$, and $\frac{\nabla_\theta \hat{\ell}(\theta^i; \widetilde{\boldsymbol{X}}, Y)}{\partial \widetilde{\boldsymbol{X}}}$ can be calculated through training. The third term $\frac{\partial \widetilde{\boldsymbol{X}}}{\partial \boldsymbol{W}}$ can be approximated during the adversarial process following the derivation below:

$$\widetilde{\boldsymbol{X}}^{i+1} = \widetilde{\boldsymbol{X}}^i + \epsilon_x \nabla_{\widetilde{\boldsymbol{X}}^i}\left\{\ell(\theta; \widetilde{\boldsymbol{X}}^i, Y) - \lambda c_w(\widetilde{\boldsymbol{X}}^i, \boldsymbol{X})\right\},$$

$$\frac{\partial \widetilde{\boldsymbol{X}}^{i+1}}{\partial \boldsymbol{W}} = \frac{\partial \widetilde{\boldsymbol{X}}^i}{\partial \boldsymbol{W}} - 2\epsilon_x \lambda \text{Diag}\left(\widetilde{\boldsymbol{X}}^i - \boldsymbol{X}\right),$$

$$\frac{\partial \widetilde{\boldsymbol{X}}}{\partial \boldsymbol{W}} \approx -2\epsilon_x \lambda \sum_i \text{Diag}(\widetilde{\boldsymbol{X}}^i - \boldsymbol{X}). \quad (11)$$

---

**Algorithm 1** Training procedure of spatiotemporal distributionally robust optimization

---

**Input:** Multi-environments EEG data $D^{e_1}, D^{e_2}, \ldots, D^{e_n}$, where $D^e = (\boldsymbol{X}^e, Y^e)$, $\boldsymbol{X}$ represents EEG signal, $Y$ represents seizure label.

**Hyperparameters:** $N, N_\theta, N_w, m, \epsilon_x, \epsilon_\theta, \epsilon_M, \alpha, \beta$

**Initialize:** $\boldsymbol{W} = \begin{bmatrix} \boldsymbol{I}_{\frac{T}{N}} \bigotimes \mathrm{diag}(\boldsymbol{W}_1 \boldsymbol{M}_1, \boldsymbol{W}_2 \boldsymbol{M}_2, \cdots, \boldsymbol{W}_N \boldsymbol{M}_N) & \boldsymbol{0}_{(C*T)\times 1} \\ \boldsymbol{0}_{1\times(C*T)} & 1 \end{bmatrix}, \boldsymbol{M}_t = \boldsymbol{I}$

**For iteration $i$ from 1 to $N$**

    **For iteration $j$ from 1 to $N_\theta - 1$**

        **Initialize $\tilde{\boldsymbol{X}}_0$ as:** $\tilde{\boldsymbol{X}}_0 = \boldsymbol{X}$

        **For iteration $k$ from 0 to $m-1$**

            `# Approximate the supreme of` $s_\lambda(\boldsymbol{X})$ `for` $\boldsymbol{X}^e$ `from all` $e \in \varepsilon$

            $\tilde{\boldsymbol{X}}_{k+1}^e = \tilde{\boldsymbol{X}}_k^e + \epsilon_x \frac{\partial(l(\theta, \tilde{\boldsymbol{X}}_k^e) - \lambda c_w(\tilde{\boldsymbol{X}}_k^e, \tilde{\boldsymbol{X}}_0^e))}{\partial \boldsymbol{X}}$.

        **End for**

        `# Update` $\theta$

        $\theta^{j+1} \leftarrow \theta^j - \epsilon_\theta \frac{\partial l(\theta_j; (\tilde{\boldsymbol{X}}_m, Y))}{\partial \theta}$.

    **End for**

    $R(\theta) = \frac{\sum_{e \in \varepsilon} \ell^e(\theta)}{|\mathcal{E}|} + \alpha \max_{p,q \in \mathcal{E}} (\ell^p - \ell^q) + \beta \sum_t \|\boldsymbol{W}_{t+1} \boldsymbol{M}_{t+1} - \boldsymbol{W}_t \boldsymbol{M}_t\|_F^2$

    $\boldsymbol{M}^{j+1} = \boldsymbol{M}^i - \epsilon_M \frac{\partial R(\theta)}{\partial \boldsymbol{M}}$.

    $\boldsymbol{M}^{j+1} = Proj_M(\boldsymbol{M}^{j+1})$

**End for.**

---

## C   Training procedure and computational cost of STDRO

The optimization process of STDRO is shown in Algorithm 1. In experiments, the time period number $N$ is set as 4, the attack iteration $m \in \{5, 10, 15, 20\}$. The product of $N$ and $N_\theta$ is set to be equal to the training epoch number of the baseline method.

In STDRO, the spatial-temporal graph is constructed in advance which hardly influences the training time. In the bi-level optimization ($\boldsymbol{W}$ and $\theta$) of our method, the total epoch number considering optimizing parameters $\theta$ is the same as the baseline, i.e. the product of $N$ and $N_\theta$ in Algorithm 1 equals the epoch number of the baseline. The additional costs primarily lie in the computation of adversarial samples (which is proportional to the iteration number $m$) and approximating second-order derivatives for stability-induced $\boldsymbol{W}$ within a single epoch. In practice, for finetuning the DCRNN model, STDRO takes about 12 minutes per epoch ($m = 10$), while the baseline takes around 3 minutes. Please note that such increase is only for the finetuning stage (50 epochs), while pretraining methods typically have much larger computational costs (44 minutes per epoch for 350 epochs). So the increase of computational costs for STDRO is not large considering the whole process.

## D   Data preprocessing details

The TUSZ V1.5.2 train dataset was randomly split into a training set and a validation set in a 90/10 ratio, consistent with previous studies (Tang et al., 2022; Afzal et al., 2024). The dataset split satisfies the cross-patient setting, i.e., the training, validation, and testing sets have different patients. We resample the EEGs into 200 Hz and split the EEG signals into 12s or 60s segments to evaluate the method's performance in both short and long-term scenarios. Other data preprocessing techniques, including whether to apply the fast Fourier transform (FFT), are consistent with our comparison baseline methods. We split each EEG segment into four time periods.

The CHB-MIT dataset was resampled from 256Hz to 64Hz. The EEG signals were split into 4-second segments. We follow the data split in previous works (Afzal et al., 2024), which selected 80% of the data for training (18 patients), 10% for evaluation (3 patients), and 10% for testing (3 patients). We split each EEG segment into four time periods.

## E HYPER-PARAMETERS SENSITIVITY ANALYSIS

We test the performance of STDRO under different environment (patient group) numbers for the stability objective in Table 7. Experiments are the seizure detection task on the TUSZ dataset with 12s time window size. The results that our approach is robust to the number of patient groups.

Table 7: Performance under different patient group numbers.

| PATIENT GROUP NUMBER | AUROC | F1-SCORE | ACCURACY | RECALL | PRECISION |
|---|---|---|---|---|---|
| K=4 | 87.9 | 55.3 | 89.1 | 61.5 | 50.1 |
| K=6 | 88.2 | 54.8 | 89.5 | 58.1 | 51.8 |

We have also conducted the sensitivity analysis of other hyperparameters, such as the number of time periods $N$, the regularization term for stability $\alpha$, and the regularization term for temporal smoothness $\beta$ .

Table 8: Performance under different values of $N$.

| $N$ | AUROC | F1-SCORE | IoU | ACCURACY | RECALL | PRECISION |
|---|---|---|---|---|---|---|
| 2 | 87.8 | 54.4 | 37.3 | 89.4 | 57.6 | 51.5 |
| 4 | 88.2 | 54.8 | 37.7 | 89.5 | 58.1 | 51.8 |
| 6 | 87.8 | 53.9 | 36.9 | 88.1 | 63.7 | 46.8 |

Table 9: Performance under different values of $\alpha$.

| $alpha$ | AUROC | F1-SCORE | IoU | ACCURACY | RECALL | PRECISION |
|---|---|---|---|---|---|---|
| 0.2 | 87.6 | 54.0 | 37.0 | 88.7 | 60.5 | 48.8 |
| 0.5 | 88.2 | 54.8 | 37.7 | 89.5 | 58.1 | 51.8 |
| 1 | 87.0 | 50.9 | 34.1 | 89.9 | 47.7 | 54.6 |
| 2 | 87.1 | 52.6 | 35.7 | 89.6 | 52.8 | 52.4 |

Table 10: Performance under different values of $\beta$.

| $\beta$ | AUROC | F1-SCORE | IoU | ACCURACY | RECALL | PRECISION |
|---|---|---|---|---|---|---|
| 2 | 87.3 | 53.3 | 36.3 | 90.0 | 52.3 | 54.3 |
| 0.2 | 87.7 | 54.5 | 37.5 | 90.0 | 55.0 | 54.1 |
| $2 \times 10^{-2}$ | 87.6 | 53.9 | 36.8 | 89.5 | 56.3 | 51.6 |
| $2 \times 10^{-4}$ | 88.2 | 54.8 | 37.7 | 89.5 | 58.1 | 51.8 |
| $2 \times 10^{-5}$ | 87.5 | 53.3 | 36.4 | 88.8 | 58.4 | 49.0 |

## F THEORETICAL ANALYSIS

In this section, we provide more theoretical analysis of the proposed method. The core idea is to leverage the manifold assumption of the data and the potential spatio-temporal uncertainty set to show that it may derive a tighter generalization bound on worst-case distributions than standard DRO. According to fundamental observation in modern machine learning of manifold hypothesis (Belkin & Niyogi, 2001; Fefferman et al., 2016), high-dimensional data tends to concentrate around a lower-dimensional manifold in the ambient space. We assume the data distribution $(x, y) \sim P$ where the inputs $x_i \in \mathcal{X} = \mathbb{R}^D$ are assumed to lie on a smooth $m$-dimensional Riemannian manifold $\mathcal{M} \subset \mathbb{R}^D$ with $m \ll D = C \times T$. For any $x \in \mathcal{M}$, $T_x\mathcal{M}$ denotes the $m$-dimensional tangent space at $x$, and $N_x\mathcal{M} = (T_x\mathcal{M})^\perp$ denotes the $(D - m)$-dimensional normal space (the orthogonal complement in $\mathbb{R}^D$). We first make the assumption that our distance metric captures such manifold.

**Assumption F.1.** There exist constants $0 < \alpha \leq \beta$ such that for every $x \in \mathcal{M}$ and any vector $v \in \mathbb{R}^D$ decomposed as $v = v^\| + v^\perp$ with $v^\| \in T_x\mathcal{M}$ and $v^\perp \in N_x\mathcal{M}$, the following inequalities

hold:

$$\sum_{t=1}^{N} v_t^\top \boldsymbol{W}_t\, v_t \;\geq\; \alpha \|v^\perp\|^2, \qquad \sum_{t=1}^{N} v_t^\top \boldsymbol{W}_t\, v_t \;\leq\; \beta \|v^\|\|^2,$$

where $\boldsymbol{W}_t \succeq 0$ is a given spatial correlation matrix for segment $t$ and $\boldsymbol{W}_t$ preserves the tangent/normal decomposition, i.e., $\boldsymbol{W}_t(T_x\mathcal{M}) \subseteq T_x\mathcal{M}$ and $\boldsymbol{W}_t(N_x\mathcal{M}) \subseteq N_x\mathcal{M}$, which means $\sum_{t=1}^{N}(v_t^\|)^\top \boldsymbol{W}_t\, v_t^\perp = 0$ for any $v^\| \in T_x\mathcal{M}$ and $v^\perp \in N_x\mathcal{M}$.

Assumption F.1 means the metric induced by $\boldsymbol{W} = \mathrm{diag}(\boldsymbol{W}_1, \ldots, \boldsymbol{W}_N)$ strongly penalizes off-manifold directions while treating on-manifold directions moderately, and that $\boldsymbol{W}$ is "block-diagonal" with respect to the $T_x\mathcal{M} \oplus N_x\mathcal{M}$ decomposition.

In DRO, an adversarial uncertainty set is typically defined via a Wasserstein ball, which requires specifying a transport cost. To simplify, we will omit 'y-part' in transport cost in the following parts. To incorporate the manifold structure, ideally, we will define a transportation cost that respects the geometry of $\mathcal{M}$:

$$c_\mathcal{M}(x, x') := \|\mathrm{Proj}_{T_x\mathcal{M}}(x' - x)\|^2 + \Lambda\,\|\mathrm{Proj}_{N_x\mathcal{M}}(x' - x)\|^2,$$

with a large $\Lambda \gg 1$ so that off-manifold displacements are extremely expensive. In practice, we approximate this with a surrogate cost $c_w$ defined using the matrices $\boldsymbol{W}_t$:

$$c_w(x, x') := \sum_{t=1}^{N}(x_t - x'_t)^\top \boldsymbol{W}_t\,(x_t - x'_t).$$

$c_w$ is a Mahalanobis cost that captures the spatial correlation structure in each time segment $x_t$, where $x_t$ denotes the $t$-th period of EEG signals $x$. Then we will prove that $c_w$ heavily penalizes off-manifold shifts and mildly penalizes on-manifold shifts, making it a tractable surrogate for $c_\mathcal{M}$.

**Lemma F.2.** *Under Assumption F.1, for any $x \in \mathcal{M}$ and any $x' \notin \mathcal{M}$, let $x^* = x + \mathrm{Proj}_{T_x\mathcal{M}}(x' - x)$, i.e., the projection of $x'$ onto the manifold at $x$. Then*

$$c_w(x, x^*) < c_w(x, x').$$

*Proof.* Let $x' - x = v^\| + u$ with $v^\| = \mathrm{Proj}_{T_x\mathcal{M}}(x' - x)$ and $u = \mathrm{Proj}_{N_x\mathcal{M}}(x' - x)$, so $x^* = x + v^\|$. If $x' \notin \mathcal{M}$ then $u \neq 0$. By definition of $c_w$ and Assumption F.1, $c_w(x, x') = \|v^\| + u\|_W^2 = \|v^\|\|_W^2 + \|u\|_W^2$, since $\langle v^\|, u\rangle_W = 0$. Meanwhile, $c_w(x, x^*) = \|v^\|\|_W^2$, so $c_w(x, x') - c_w(x, x^*) = \|u\|_W^2$. Assumption F.1 ensures $\|u\|_W^2 \geq \alpha\|u\|^2 > 0$. Thus $c_w(x, x') - c_w(x, x^*) > 0$, i.e., $c_w(x, x') > c_w(x, x^*)$. $\square$

This means moving $x$ to the on-manifold point $x^*$ costs strictly less in $c_w$ than moving it the same Euclidean distance in an off-manifold direction towards $x'$. This implies any optimal transport plan under cost $c_w$ will never choose an off-manifold target if an on-manifold alternative exists.

**Spatiotemporal Uncertainty Set.** We define the STDRO adversarial uncertainty set as the Wasserstein ball around $P_0$ using cost $c_w$:

$$\mathcal{H}_{\mathrm{ST}} := \{\, Q : W_{c_w}(Q, P_0) \leq \rho \,\}.$$

Here $W_{c_w}(Q, P_0)$ denotes the Wasserstein distance with transportation cost $c_w$. In contrast, the isotropic DRO would use Wasserstein distance with Euclidean cost $c_0(x, x') = \|x - x'\|^2$, defining

$$\mathcal{H}_{\mathrm{iso}} := \{\, Q : W_{c_0}(Q, P_0) \leq \rho \,\}.$$

By Lemma F.2, any optimal transport plan for $W_{c_w}(P_0, Q)$ can be adjusted to remain on $\mathcal{M}$ without increasing cost. If in addition the loss function does not vary , then the adversary has little incentive to leave $\mathcal{M}$. The next proposition makes this precise:

**Proposition F.3** (Near-Manifold Worst-Case Distribution)**.** *Under Assumption 1 and assuming the loss $\ell(\theta; x, y)$ is L-Lipschitz continuous in $x$ along normal directions, the worst-case risk over $\mathcal{H}_{ST}$ can be approximated by a distribution supported on $\mathcal{M}$ with small errors. Specifically, for any $Q \in \mathcal{H}_{ST}$, there exists $Q' \in \mathcal{H}_{ST}$ with $\mathrm{supp}(Q') \subset \mathcal{M}$ such that*

$$\sup_{\theta \in \Theta} \left| \mathbb{E}_{(x,y)\sim Q}[\ell(\theta; x, y)] - \mathbb{E}_{(x,y)\sim Q'}[\ell(\theta; x, y)] \right| \leq L\sqrt{\frac{\rho}{\alpha}}\,.$$

*Proof.* For any $Q \in \mathcal{H}_{\mathrm{ST}}$, construct a new distribution $Q'$ by moving each point $z = (x, y)$ in $Q$ to $z^* = (x^*, y)$ where $x^* = x + \mathrm{Proj}_{T_{x_0}\mathcal{M}}(x - x_0)$ and $x_0 \in \mathcal{M}$ is the source point in $P_0$ matched to $z$ under $W_{c_w}(P_0, Q) \leq \rho$. By Lemma F.2, the cost of $W_{c_w}(P_0, Q)$ does not increase under this modification, so $W_{c_w}(P_0, Q') \leq \rho$ (i.e., $Q' \in \mathcal{H}_{\mathrm{ST}}$) and $\mathrm{supp}(Q') \subset \mathcal{M}$. For each hypothesis $\theta$,

$$\left| \mathbb{E}_{(x,y)\sim Q}[\ell(\theta; x, y)] - \mathbb{E}_{(x,y)\sim Q'}[\ell(\theta; x, y)] \right| \leq L\, \mathbb{E}_{(x,y)\sim Q}\left[ \|x - \mathrm{Proj}_{\mathcal{M}}(x)\| \right],$$

since the loss is $L$-lipchitz continuous in $x$ along normal directions. Note that for each $(x, y)$ in $\mathrm{supp}(Q)$, $c_w((x_0, y), (x, y)) \geq \alpha \|x - x^*\|^2$ by Assumption F.1. Thus $\|x - x^*\|^2 \leq \frac{1}{\alpha} c_w((x_0, y), (x, y))$. Taking expectation gives $\mathbb{E}_{(x,y)\sim Q}\|x - x^*\|^2 \leq \frac{1}{\alpha} W_{c_w}(P_0, Q) \leq \frac{\rho}{\alpha}$. By Cauchy-Schwarz, $\mathbb{E}\|x - x^*\| \leq \sqrt{\frac{\rho}{\alpha}}$. Therefore the loss difference above is bounded by $L\sqrt{\frac{\rho}{\alpha}}$, uniformly in $\theta$. Taking supremum over $\theta$ yields the claimed inequality. $\square$

**Generalization Bound.** Finally, we show that if the worst-case distribution is preliminarily constrained to a manifold, the distributionally robust generalization guarantee bound will be tighter than the standard DRO. Let $\{x_i, y_i\}_{i=1}^n \sim P_0$ constitute the empirical distribution $P_n$. Denote the true worst-case risk as $R_{\mathrm{ST}}(\theta) = \sup_{Q \in \mathcal{H}_{\mathrm{ST}}} \mathbb{E}_{(x,y)\sim Q}[\ell(\theta; x, y)]$, and empirical worst-case risk as $R_n(\theta) = \sup_{Q:W_{c_w}(Q, P_n)\leq \rho} \mathbb{E}_{(x,y)\sim Q}[\ell(\theta; x, y)]$. Define $\hat{\theta} = \arg\min_\theta R_n(\theta)$.

**Theorem F.4** (Generalization Bounds). *Assume $|\ell(\theta; x, y)| \leq 1$ for all $(\theta, x, y)$, $\|x\| \leq r$, and $\ell(\theta; x, y)$ is $L$-lipschitz continuous in $x$ for every $\theta \in \Theta$. For any $0 < \delta < 1$, with probability at least $1 - \delta$, we have*

$$\sup_{Q \in \mathcal{H}_{ST}} \mathbb{E}_{(x,y)\sim Q}[\ell(\hat{\theta}; x, y)] \leq \sup_{Q:W_{c_w}(Q, P_n)\leq \rho} \mathbb{E}_{(x,y)\sim Q}[\ell(\hat{\theta}; x, y)] + \mathfrak{R}_n(\mathcal{F}_{adv}) + \sqrt{\frac{\ln(1/\delta)}{2n}},$$

*where $\mathfrak{R}_n(\mathcal{F}_{adv})$ is the Rademacher complexity of the adversarial loss class*

$$\mathcal{F}_{adv} := \left\{ (x, y) \mapsto \sup_{Q \in \mathcal{H}_{ST}} \mathbb{E}_{(x',y)\sim Q}[\ell(\theta; x', y)] \; : \; \theta \in \Theta \right\}.$$

*When the worst-case distributions are constrained on the manifold, we have*

$$\mathfrak{R}_n(\mathcal{F}_{adv}) = \mathcal{O}\left(\sqrt{\frac{m}{n}}\right),$$

*which is smaller than the standard DRO complexity bound $\mathcal{O}\left(\sqrt{\frac{D}{n}}\right)$ given $m \ll D$.*

*Proof.* We apply a uniform convergence bound based on Rademacher complexity. Applying the standard result of Bartlett & Mendelson (2002); Sinha et al. (2018); Wainwright (2019), for any fixed $\theta$ and $\delta$, with probability $1 - \delta$:

$$\sup_{\theta \in \Theta} |R_{\mathrm{ST}}(\theta) - R_n(\theta)| \leq \mathfrak{R}_n(\mathcal{F}_{adv}) + \sqrt{\frac{\ln(1/\delta)}{2n}}.$$

Thus we get the stated generalization bound.

Then we estimate the order of $\mathfrak{R}_n(\mathcal{F}_{adv})$. We use Dudley's entropy integral:

$$\mathfrak{R}_n(\mathcal{F}_{adv}) \leq \frac{12}{\sqrt{n}} \int_0^1 \sqrt{\ln N(\varepsilon, \mathcal{F}_{adv}, L_2(P_n))}\, d\varepsilon,$$

where $N(\varepsilon, \mathcal{F}, L_2(P_n))$ is the covering number of $\mathcal{F}$ at radius $\varepsilon$ under the empirical $L_2$ norm. Since $l(\theta; x, y)$ is $L$-lipschitz, we have:

$$N(\varepsilon, \mathcal{F}, L_2(P_n)) \leq N(\varepsilon, \mathcal{F}, L_\infty(S)) \leq N\left(\frac{\varepsilon}{L}, \mathcal{X}, \|\cdot\|\right), \tag{12}$$

where $L_\infty(S)$ is the norm for samples and $N(\epsilon, \mathcal{X}, \|\cdot\|)$ is the $\epsilon$-covering number for the input space. When the worst-case distributions are constrained on the manifold, we have $N\left(\frac{\varepsilon}{L}, \mathcal{X}, \|\cdot\|\right) \leq \left(\frac{cLr}{\varepsilon}\right)^m$, therefore:

$$\mathfrak{R}_n(\mathcal{F}_{adv}) \leq \frac{12}{\sqrt{n}} \int_0^1 \sqrt{\ln N(\varepsilon, \mathcal{F}_{adv}, L_2(P_n))}\, d\varepsilon \leq \mathcal{O}\left(\sqrt{\frac{m}{n}} \int_0^1 \sqrt{\ln \frac{1}{\varepsilon}}\, d\varepsilon\right) = \mathcal{O}\left(\sqrt{\frac{m}{n}}\right),$$

while for the standard DRO with input dimension as $D$, the complexity bound is correspondingly $\mathcal{O}\left(\sqrt{\frac{D}{n}}\right)$.

$\square$

