# OpenReview forum: "Spatiotemporal distributionally robust optimization for improved cross-patient EEG seizure analysis"
_ICLR.cc/2026/Conference — Submitted to ICLR 2026_

### Official Review · Reviewer_fEhV · 2025-10-25

**Soundness:** 1
**Presentation:** 2
**Contribution:** 2
**Rating:** 2
**Confidence:** 5

**Summary:**

This paper introduces STDRO, a spatiotemporal distributionally robust optimization framework for cross-patient EEG seizure detection and classification.

The proposed method enhances existing architectures (DCRNN, GraphS4former, VQ-MTM) with a new optimization objective that learns uncertainty sets capturing EEG spatial correlations and temporal dependencies. These uncertainty sets are parameterized by learnable matrices and regularized through a “stability objective” designed to reduce performance gaps between patient groups.

STDRO is presented as a plug-and-play training strategy applicable to any architecture, and it demonstrates moderate improvements on the TUSZ and CHB-MIT datasets.

**Strengths:**

- The Introduction and Related Work sections are well written and supported by numerous references covering the various aspects discussed in the paper.


- Topic Relevance: Cross-patient generalization is a fundamental and challenging problem in EEG analysis, and addressing it from an optimization rather than an architectural or pretraining perspective is an interesting direction.

**Weaknesses:**

- Figure 1: The figure lacks visual clarity, particularly the box in the bottom-right corner, which looks messy.
- Optimization complexity: The method substantially alters the optimization problem, making it potentially much harder to tune. There is no analysis or evidence illustrating how this affects training stability, convergence, or runtime. The added bi-level optimization seems to introduce major complexity for relatively small empirical gains.
- Objective formulation: The objective remains vague and underspecified. The definition of the stability term and patient grouping strategy appears arbitrary. The learnable matrix Mₜ is poorly explained. Its structure and dimensionality are unclear, and there is no intuition for what it learns or how it interacts with Wₜ.
- Inconsistent improvements: There is no explanation for the large variability in reported gains (e.g., from +58% to +69% in one setup but only +0–5% in others).
- Statistical significance: The lack of significance testing is a serious issue. Given the small differences in performance, it’s entirely possible that the improvements are not meaningful. Ablation studies should include multiple seeds or data splits to assess robustness.
- Splitting protocol: TUSZ has an official split, yet the authors use a 90/10 random split, breaking the standard evaluation protocol and compromising comparability. CHB-MIT lacks a predefined split, so proper cross-validation is expected but not described.
- Visualization: The covariate-weight figures are illustrative but fail to demonstrate how or why STDRO improves robustness.
- Presentation: The paper is algebraically overloaded. Core ideas should be kept concise in the main text, with derivations moved to the appendix.
- Generality: The method is heavily tailored to seizure datasets, suggesting limited transferability to other time-series tasks without further validation.
- Metrics: The authors only report sample-based metrics (F1, AUROC), while the field is increasingly shifting toward event-based metrics (IoU, IoU@0.5) that better capture the real objective of seizure detection. Reporting F1 for comparison is fine, but including event-level metrics would have provided a more complete picture.

**Questions:**

1. Code release: Since STDRO is designed as a plug-and-play optimization layer, do you plan to release the implementation publicly to facilitate reproducibility and adoption?

2. Statistical validation: Could you include statistical significance tests (e.g., across seeds or splits) in the ablation and main results to confirm that the observed improvements are robust and meaningful?

3. Training behavior: Can you provide more insight into the practical impact of the proposed optimization on training dynamics (compute time, stability, and convergence) compared to standard baselines?

4. Hyperparameter sensitivity: How sensitive are your results to the choice of α and β in Eq. (2)? Some intuition or plots showing their influence would clarify how to tune the method.

5. Interpretability of Mₜ: Could you offer a more intuitive explanation or visualization of what the learnable matrix Mₜ actually learns and how it interacts with the fixed Wₜ during training?

6. Split protocol and reproducibility: For TUSZ, the 90/10 random split deviates from the predefined standard. Or maybe you use it? Could you clarify how you ensured patient disjointness and comparability with prior work?

7. Generalization beyond seizure data: Have you tried (or could you speculate on) applying STDRO to other time series (other EEG datasets like BCI tasks or other signals like ECG) to test its claimed generality?

---

> ### Author Response · Authors · 2025-11-26
> **Response to Reviewer fEhV (Part 1)**
>
> Thank you for your review and valuable feedback. We carefully address your concerns as follows.
>
> 1. Presentation of Figure 1.
>
> We have revised the box in the bottom-right corner. In this part, we want to show the alternating optimization of model parameter and uncertainty set. On one hand, through robust optimization on the uncertainty set, model parameters are updated. On the other hand, give EEG inputs, model will output the performance of different patient groups, based on which we can calculate the stability function consisting of average performance and max margin performance. Through gradient backpropagation of stability function, the information to update uncertainty set is obtained.
>
> 2. Optimization complexity.
>
> The training stability and convergence is similar to the baseline methods. As for the runtime, our total epoch number considering optimizing parameters $\theta$ is the same as the baseline (the product of $N$ and $N_{\theta}$ in Algorithm 1 equals the epoch number of the baseline). The additional costs primarily lie in the computation of adversarial samples (which is proportional to the iteration number $m$) and approximating second-order derivatives for stability-induced $W$ within a single epoch. In practice, for finetuning the DCRNN model, STDRO takes about 12 minutes per epoch ($m=10$), while the baseline takes around 3 minutes. However, please note that such increase is only for the finetuning stage (50 epochs), while pretraining methods typically have much larger computational costs (44 minutes per epoch for 250 epochs). So the increase of computational costs for STDRO is not large considering the whole process.
>
> We respectively disagree that our method introduces major complexity for relatively small empirical gains. First, as explained above, the overhead of our method is not large compared to other methods such as introducing a pretraining stage. Second, the empirical gains are consistent and not small, which is also complementary to existing methods.
>
> 3. Objective formulation.
>
> Our objective formulation is primarily based on the Distributionally Robust Optimization (DRO) formulation [1-3] (described in Section 3.1.1), which is a min-max optimization problem to optimize parameters over the worst-case distributions in the uncertainty set. The major difference is that we construct the uncertainty set in DRO based on the transportation cost function described in Eq. (2), whose covariate weight will also be updated based on a stability objective. This leads to our overall objective described in Eq. (3).
>
> 4. About stability and patient grouping strategy
>
> The definition of the stability term is borrowed from the established causal invariant learning [4-6], which aims to distinguish stable and unstable features through model performance’s robustness across different environments (groups). Our group strategy was primarily adopted for implementation simplicity. It could be further refined through iterative optimization [7], which we leave as future work.
>
>
> 5. Explanation of the learnable matrix $M_t$.
>
> The matrix $M_t \in R^{C\times C}$ is used to parameterize the uncertainty set so that it can be optimized for the stability objective. In the transportation cost function, $W_t M_t$ is equivalent to the symmetric matrix $0.5* (W_t M_t + M_t' W_t’) $ since $c W_t M_t c'=c \frac{1}{2} (W_t M_t + M_t' W_t’) c'$. So it interacts with $W$ by adjusting the ellipse induced by the covariant matrix. $M_t$ is initialized to be identity matrix so that the uncertainty set driven from $W_tM_t$ is initialized based on spatiotemporal structures; further, the uncertainty set can be adjusted by updating $M_t$ for the stability objective (Eq. (3)). This approach enables the simultaneous consideration of both spatiotemporal and stability objectives.
>
> 6. About the large variability in reported gains.
>
> The extent of improvement is method-dependent, and there exists variations in results across different approaches, tasks and metrics. We also observe inconsistent gains across tasks and metrics in previous works, and we do not expect a uniform gain.

---

> ### Author Response · Authors · 2025-11-26
> **Response to Reviewer fEhV (Part 2)**
>
> 7. Multiple runs in ablation studies.
>
> We have updated the results of ablation studies with multiple runs under different random seeds (Table 6, quoted below). The results also support the effectiveness of DRO and each component in our STDRO, showing that the improvements are robust and meaningful.
>
> **Table 1. Ablation study on 12s cross-patient seizure detection on the TUSZ dataset.**
>
> | Method | **AUROC** | **F1-score** | Accuracy | Recall | Precision |
> | :--- | ---: | ---: | ---: | ---: | ---: |
> | DCRNN | $0.865 \pm 0.010$ | $0.495 \pm 0.013$ | $0.868 \pm 0.013$ | $0.589 \pm 0.030$ | $0.429 \pm 0.031$ |
> | +DRO | $0.868 \pm 0.006$ | $0.514 \pm 0.001$ | $0.883 \pm 0.003$ | $0.564 \pm 0.009$ | $0.472 \pm 0.010$ |
> | +DRO w/ spatiotemporal W | $0.874 \pm 0.005$ | $0.524 \pm 0.007$ | $0.878 \pm 0.001$ | $0.612 \pm 0.050$ | $0.462 \pm 0.036$ |
> | +DRO w/ stability-induced W | $0.875 \pm 0.013$ | $0.521 \pm 0.034$ | $0.881 \pm 0.026$ | $0.585 \pm 0.077$ | $0.486 \pm 0.101$ |
> | +STDRO (ours) | $0.876 \pm 0.008$ | $0.542 \pm 0.007$ | $0.893 \pm 0.002$ | $0.570 \pm 0.008$ | $0.510 \pm 0.008$ |
>
> 8. Splitting protocol
>
> We follow the protocol in previous works. As for the splitting protocol of TUSZ, we use exactly the same split as in [8,9]. Note that we used TUSZ V1.5.2 which does not have the train-validation split, although the latest version of TUSZ has a split. We choose this version following the baseline methods DCRNN [8] and graphS4former [9].
>
> As for the split of CHB-MIT, we follow the description in previous works [10], which selected 80% of the patients for training (18 patients), 10% for evaluation (3 patients), and 10% for testing (3 patients). We will add the description in the paper.
>
> 9. About the covariate-weight figures and robustness.
>
> The covariate-weight figures mainly aim to illustrate that our stability component effectively influences the uncertainty set structure through time.
>
> The improvement in robustness is supported by the theoretical advantage of DRO and STDRO. Firstly, DRO is defined to improve robustness by optimizing over the worst-case distribution in the uncertainty set, ensuring good performance not only at the training distribution but also across neighborhood regions with shifts. When the test set has distribution shift (e.g., cross-patient), DRO can lead to better performance. Theoretically, empirical risk minimization (train on the training distribution) cannot guarantee generalization on shifted distributions, while DRO has the generalization bound on the shifted neighborhood distribution [2]. Secondly, we have further supplemented theoretical analysis of STDRO in Appendix F, showing that it may tighten the generalization bound under the manifold assumption of the data compared to standard DRO.
>
> 10. About the presentation.
>
> Thank you for the valuable feedback. We will revise the paper to better highlight the core idea.
>
> 11. Transferability to other time-series tasks.
>
> We add new experiments of an EEG decoding task (sleep stage classification) on the DOD-H dataset. We apply STDRO to the state-of-the-art method GraphS4former. The experimental results with multiple runs also verified the effectiveness of our method.
>
> **Table 2. Results of cross-subject sleeping stage classification on the DOD-H dataset.**
>
> | Method | **Macro-F1** | **Kappa** |
> |--------|--------------|-----------|
> | LSTM | 0.609 $\pm$ 0.034 | 0.539 $\pm$ 0.046 |
> | SimpleSleepNet | 0.720 $\pm$ 0.001 | 0.703 $\pm$ 0.013 |
> | RobustSleepNet | 0.777 $\pm$ 0.007 | 0.758 $\pm$ 0.008 |
> | DeepSleepNet | 0.716 $\pm$ 0.025 | 0.711 $\pm$ 0.032 |
> | GraphS4former | 0.810 $\pm$ 0.015 | 0.790 $\pm$ 0.020 |
> | GraphS4former+STDRO (ours) | **0.822 $\pm$ 0.011** | **0.807 $\pm$ 0.010** |

---

> > ### Author Response · Authors · 2025-11-26
> > **Response to Reviewer fEhV (Part 3)**
> >
> > 12. About the metric IoU and hyperparameter sensitivity.
> >
> > We add the new IoU metric in the newly added parameter sensitivity analysis experiments. Specifically, we study the sensitivity analysis of parameter $\alpha$, $\beta$ and $N$, and simultaneously report IoU.
> >
> > **Table 3: Performance under different values of $N$**
> >
> > | $N$ | **AUROC** | **F1-score** | **IoU** | Accuracy | Recall | Precision |
> > |-----|-----------|--------------|---------|----------|--------|-----------|
> > | 2   | 87.8      | 54.4         | 37.3    | 89.4     | 57.6   | 51.5      |
> > | 4   | 88.2      | 54.8         | 37.7    | 89.5     | 58.1   | 51.8      |
> > | 6   | 87.8      | 53.9         | 36.9    | 88.1     | 63.7   | 46.8      |
> >
> > **Table 4: Performance under different values of $\alpha$**
> >
> > | $\alpha$ | **AUROC** | **F1-score** | **IoU** | Accuracy | Recall | Precision |
> > |----------|-----------|--------------|---------|----------|--------|-----------|
> > | 0.2      | 87.6      | 54.0         | 37.0    | 88.7     | 60.5   | 48.8      |
> > | 0.5      | 88.2      | 54.8         | 37.7    | 89.5     | 58.1   | 51.8      |
> > | 1        | 87.0      | 50.9         | 34.1    | 89.9     | 47.7   | 54.6      |
> > | 2        | 87.1      | 52.6         | 35.7    | 89.6     | 52.8   | 52.4      |
> >
> > **Table 5: Performance under different values of $\beta$**
> >
> > | $\beta$ | **AUROC** | **F1-score** | **IoU** | Accuracy | Recall | Precision |
> > |---------|-----------|--------------|---------|----------|--------|-----------|
> > | 2       | 87.3      | 53.3         | 36.3    | 90.0     | 52.3   | 54.3      |
> > | 0.2     | 87.7      | 54.5         | 37.5    | 90.0     | 55.0   | 54.1      |
> > | $2 \times 10^{-2}$ | 87.6 | 53.9         | 36.8    | 89.5     | 56.3   | 51.6      |
> > | $2 \times 10^{-4}$ | 88.2 | 54.8         | 37.7    | 89.5     | 58.1   | 51.8      |
> > | $2 \times 10^{-5}$ | 87.5 | 53.3         | 36.4    | 88.8     | 58.4   | 49.0      |
> >
> > 13. Code release.
> >
> > We plan to release the code upon the acceptance.
> >
> > [1] Michel P, et al. Modeling the Second Player in Distributionally Robust Optimization. ICLR 2021.
> >
> > [2] Sinha A, et al. Certifying Some Distributional Robustness with Principled Adversarial Training. ICLR 2018.
> >
> > [3] Staib M and Jegelka S. Distributionally robust optimization and generalization in kernel methods. NeurIPS 2019.
> >
> > [4] Liu J, et al. Heterogeneous risk minimization. ICML 2021.
> >
> > [5] Kuang K, et al. Stable prediction with model misspecification and agnostic distribution shift. AAAI 2020.
> >
> > [6] Kuang K, et al. Stable prediction across unknown environments. SIGKDD 2018.
> >
> > [7] Liu J, et al. Heterogeneous risk minimization. ICML 2021.
> >
> > [8] Tang S, et al. Self-Supervised Graph Neural Networks for Improved Electroencephalographic Seizure Analysis. ICLR 2022.
> >
> > [9] Tang S, et al. Modeling multivariate biosignals with graph neural networks and structured state space models. CHIL 2023.
> >
> > [10] Afzal A, et al. REST: Efficient and Accelerated EEG Seizure Analysis through Residual State Updates. ICML 2024.

---

### Official Review · Reviewer_n1TW · 2025-11-04

**Soundness:** 1
**Presentation:** 2
**Contribution:** 1
**Rating:** 0
**Confidence:** 5

**Summary:**

This paper provides a highly complex optimisation approach for EEG data, which is plug-&-play for existing EEG-seizure datasets and architectures. This is a new take, because prior work usually looked at this problem from an architecture perspective. It is supposed to do temporal localisation of EEG to find when in the multichannel stream a seizure happened and also classify the seizure type.

**Strengths:**

Gives great results and beats SOTA.

**Weaknesses:**

Unfortunately, the paper is not well organized. There is too much nomenclature, and it just feels like there is a lack of focus. The more one reads, the more stuff is discovered: uncertainty sets, dynamics, spatial correlations, spatio-temporal constraints, stability, adversarial optimisation, bilevel optimisation with relaxation, multi-environment objectives, wassertian distance, graph learning, transportation cost functions etc. All of this makes for a very confusing read. Because of this, I believe that the paper is very incoherent and not suitable for publication at the moment.

**Questions:**

What do they mean by ‘spatial correlations’? Are they referring to the use of multi-channel EEG information? How is that different from ‘dimensional variance’? It feels like the words channel, dimensions, and spatial are used interchangeably.

---

> ### Author Response · Authors · 2025-11-26
> **Response to Reviewer n1TW (Part 1)**
>
> We respectfully disagree with the reviewer's point that the paper is incoherent. We use common nomenclature as in the literature, and our focus is clear: adopting and improving distributionally robust optimization (DRO) for cross-patient EEG seizure analysis.
>
> 1. Terminology in DRO and EEG analysis literature.
>
> In the paper, we used standard and commonly-used terminology in DRO and EEG seizure analysis. We give more explanations of specific terms below:
>
> (1)	Uncertainty set: A collection of probability distributions around the empirical distribution. DRO optimizes against the worst-case distribution within this set [1,2,3]. An adequate choice of the uncertainty set is critical to the success of DRO [1,3,4]. It is a common term in DRO.
>
> (2)	Adversarial Optimization: To optimize the worst-case cost over an uncertainty set, adversarial attack is performed independently on each data point within a ball around itself to maximize the loss of current classification model [5,6,7].  Adversarial Optimization refers to a minimax framework with outer minimization over model parameters, inner maximization over distributions in the uncertainty set [5,6,7]. It is a common term in DRO.
>
> (3)	Wasserstein Distance and transportation cost function：The DRO uncertainty set has typically been defined as a Wasserstein ball around the empirical distribution [5,8]. Wasserstein distances define a notion of closeness between distributions, and the definition of Wasserstein distance (relying on a transportation cost function) is in Line 194-195. These are common terms in machine learning and statistics.
>
> (4)	Bilevel Optimization: Bilevel optimization is optimization problems with a hierarchical structure, involving two levels of optimization tasks, where one task is nested inside the other [9]. The adversarial optimization process of DRO is a bilevel optimization problem, and the uncertainty set optimization is also bilevel (see the objective in Eq. (3), upper-level problem is to solve the model parameters $\theta$, lower-level problem is to solve uncertainty set parameters $M$). This is a common term in optimization and machine learning.
>
> (5)	Dynamics, spatial correlations, graph：These terms follow the established usage in the EEG seizure analysis domain [10,11,12], such as, “electroencephalogram (EEG) measures the dynamics of the electrical activity of the brain in a non-invasive way [11]”, “dynamic connectivity [10]”, “EEG sensors have complex, non-Euclidean spatial correlations [12]”,”Modeling spatiotemporal dependencies in multivariate biosignals is challenging due to complex spatial correlations between the electrodes.[10,12]” DCRNN [10] is a classic graph-based modeling approach for EEG-based seizure detection and classification.
>
> (6)	Stability and Environment: These are common concepts in robust learning. There exist stable features and unstable features which can be distinguished through model performance’s robustness against different environments. Taking the animal classification as an example, the background (such as grass or water) is the unstable feature, while the animal (such as cow) in the picture is the stable feature. Stability and Environment are commonly-used terms in causal invariant learning [13-15].
>
> (7)	Spatio-temporal constraints: This is the term in our method that describes the structure of EEG data for constructing uncertainty sets in the DRO scenario.
>
> 2. Spatial correlation and EEG channels.
>
> The EEG channels have spatial distributions in the brain, so in many previous literature, “spatial” refers to EEG channels [10,12], specifically emphasizing its non-Euclidean spatial structure. The “dimension” is defined as channels multiplied by time, i.e., the data dimension. We will add explanations in the paper.

---

> > ### Author Response · Authors · 2025-11-26
> > **Response to Reviewer n1TW (Part 2)**
> >
> > [1] Michel P, et al. Modeling the Second Player in Distributionally Robust Optimization. ICLR 2021.
> >
> > [2] Hu Z and Hong L J. Kullback-Leibler divergence constrained distributionally robust optimization. Available at Optimization Online, 2013.
> >
> > [3] Levy D, et al. Large-scale methods for distributionally robust optimization. NeurIPS 2020.
> >
> > [4] Liu J, et al. Distributionally robust optimization with data geometry. NeurIPS 2020.
> >
> > [5] Sinha A, et al. Certifying Some Distributional Robustness with Principled Adversarial Training. ICLR 2018.
> >
> > [6] Duchi J C and Namkoong H. Learning models with uniform performance via distributionally robust optimization. The Annals of Statistics, 2021.
> >
> > [7] Mohajerin Esfahani P and Kuhn D. Data-driven distributionally robust optimization using the Wasserstein metric: Performance guarantees and tractable reformulations. Mathematical Programming, 2018.
> >
> > [8] Staib M and Jegelka S. Distributionally robust optimization and generalization in kernel methods. NeurIPS 2019.
> >
> > [9] Liu R, et al. Investigating bi-level optimization for learning and vision from a unified perspective: A survey and beyond. TPAMI, 2021.
> >
> > [10] Tang S, et al. Self-Supervised Graph Neural Networks for Improved Electroencephalographic Seizure Analysis. ICLR 2022.
> >
> > [11] Gui H, et al. Vector quantization pretraining for eeg time series with random projection and phase alignment. ICML 2024.
> >
> > [12] Tang S, et al. Modeling multivariate biosignals with graph neural networks and structured state space models. CHIL 2023.
> >
> > [13] Liu J, et al. Heterogeneous risk minimization. ICML 2021.
> >
> > [14] Kuang K, et al. Stable prediction with model misspecification and agnostic distribution shift. AAAI 2020.
> >
> > [15] Kuang K, et al. Stable prediction across unknown environments. SIGKDD 2018.

---

### Official Review · Reviewer_hGu2 · 2025-11-06

**Soundness:** 3
**Presentation:** 3
**Contribution:** 3
**Rating:** 8
**Confidence:** 3

**Summary:**

This paper proposes a Distributionally Robust Optimization (DRO) framework that incorporates EEG spatio-temporal structure and patient-group-wise stability to learn the uncertainty set from training data. This framework is proposed in the context of seizure detection and classification, aiming to gain generalization across patients. Optimization for the seizure detection/classification model and the data perturbation are done alternatively. The proposed framework when combined with different models show consistent performance improvement on 2 seizure classification/detection benchmarks TUSZ and CHB-MIT.

**Strengths:**

The idea of using DRO on EEG to improve cross-patient generalization is sensible and, to my best knowledge, is new in the EEG domain.

The proposal to incorporate the EEG-specific constraints related to spatiotemporal structure and group-wise stability is interesting as well, showing the importance of these domain-knowledge in modelling. The bi-level optimization problem framing is reasonable.

The evaluation results obtained by combining the proposed standalone framework with multiple existing models leading to consistent performance improvements are convincing. The ablation study further supports the technical contributions in the work.

**Weaknesses:**

It lacks some analyses that would be interesting to shed more light on what happens underneath. For example, looking at a confusion matrix with and without STDRO to understand what the model does by learning on the worst case scenario vs the average scenario.

Using average EEG signals to cluster patients appears to be suboptimal. It would be interesting to see how the groups formed by this clustering align with clinical/physiological/demographic characteristics of the patients.

It only uses seizure detection and classification to demonstrate the advantage of the proposed framework. I believe it would be useful for other EEG tasks as well and evaluation on more EEG tasks would showcase the generalization of the framework.

Even though the framework is plug-and-play, there are no analyses on the complexity and computational cost. Particularly it involves bi-level optimization and building spatio-temporal graphs.

**Questions:**

- In seizure classification performance, the framework seems to trade precision for recall. Is this an expected result by modeling based on the uncertainty set learned by the framework? Could you comment more about this?

- How can we interpret the resulting uncertainty set in terms of the distributional shift from the average scenario, making learning from it leads to better performance on the test set? Does it also mean that the model trained on the uncertainty set works less well on the average training subjects?

- Could you comment on the sensitivity of hyperparameters like number of time periods, regularization terms for stability and temporal smoothness, etc.?

---

> ### Author Response · Authors · 2025-11-26
> **Response to Reviewer hGu2 (Part 1)**
>
> We sincerely appreciate your recognition of our work’s merit and your insightful suggestions for improvement. We carefully address your concerns as follows.
>
> 1. Confusion matrix with and without STDRO.
>
> Thank you for your suggestion. We provide the confusion matrix in the newly added Figure 3B. As shown in the Figure, STDRO can improve the accuracy of the worst class (GN, from 0.053 to 0.149), whose improvement is larger than those of other classes.
>
> 2. Patient groups and grouping strategy.
>
> We visualize the seizure clinical characteristics of different groups. As shown in the newly added Figure 3A, the groups formed by EEG clustering has different seizure proportion, indicating that the grouping partially capture some features. Our grouping strategy was primarily adopted for implementation simplicity. It could be further refined through iterative optimization [1], which we leave as future work.
>
> 3. Generalization to other time-series tasks.
>
> We add a new experiment of EEG decoding task on the DOD-H dataset (sleep stage classification). We apply STDRO to the state-of-the-art method GraphS4former. The experimental results also verified the effectiveness of our method.
>
> **Table 1. Results of cross-subject sleeping stage classification on the DOD-H dataset.**
>
> | Method | **Macro-F1** | **Kappa** |
> |--------|--------------|-----------|
> | LSTM | 0.609 $\pm$ 0.034 | 0.539 $\pm$ 0.046 |
> | SimpleSleepNet | 0.720 $\pm$ 0.001 | 0.703 $\pm$ 0.013 |
> | RobustSleepNet | 0.777 $\pm$ 0.007 | 0.758 $\pm$ 0.008 |
> | DeepSleepNet | 0.716 $\pm$ 0.025 | 0.711 $\pm$ 0.032 |
> | GraphS4former | 0.810 $\pm$ 0.015 | 0.790 $\pm$ 0.020 |
> | GraphS4former+STDRO (ours) | **0.822 $\pm$ 0.011** | **0.807 $\pm$ 0.010** |
>
> 4. Complexity and computational cost.
>
> We provide more discussion on the computational costs. The spatial-temporal graph is constructed in advance which hardly influences the training time. In the bi-level optimization ($W$ and $\theta$) of our method, the total epoch number considering optimizing parameters $\theta$ is the same as the baseline (the product of $N$ and $N_{\theta}$ in Algorithm 1 equals the epoch number of the baseline). The additional costs primarily lie in the computation of adversarial samples (which is proportional to the iteration number $m$) and approximating second-order derivatives for stability-induced $W$ within a single epoch. In practice, for finetuning the DCRNN model, STDRO takes about 12 minutes per epoch ($m=10$), while the baseline takes around 3 minutes. Please note that such increase is only for the finetuning stage (50 epochs), while pretraining methods typically have much larger computational costs (44 minutes per epoch for 250 epochs). So the increase of computational costs for STDRO is not large considering the whole process.
>
> 5. Tradeoff between precision and recall.
>
> In practice, precision and recall often trade off against each other, meaning when one is high, the other may be low. So previous works mainly focus on the F1-score as a suitable metric. This fluctuation of precision and recall could be due to random variation rather than a trend consistently and intentionally caused by the method.
>
> 6. About the uncertainty set and training distribution.
>
> In DRO, the uncertainty set defines a neighborhood around the input with perturbations. The objective is to optimize for the worst-case scenario within this neighborhood, ensuring good performance not only at the training distribution but also across neighborhood regions with shifts. When the test set has distribution shift (e.g., cross-patient), DRO can lead to better performance. Theoretically, empirical risk minimization (train on the training distribution) cannot guarantee generalization on shifted distributions, while DRO has the generalization bound on the shifted neighborhood distribution [2]; and we further supplement a theoretical analysis in Appendix F to show that our proposed STDRO may have a tighter generalization bound compared to standard DRO. On the other hand, considering the performance on i.i.d data, there can be a trade-off between robustness and accuracy [3, 4] which is commonly recognized in the literature.

---

> > ### Author Response · Authors · 2025-11-26
> > **Response to Reviewer hGu2 (Part 2)**
> >
> > 7. Sensitivity analysis.
> >
> > We have added the sensitivity analysis of parameters such as regularization terms for stability $\alpha$, regularization terms for stability and temporal smoothness $\beta$ and number of time periods $N$ in Appendix E, and the results are quoted below.
> >
> > **Table 2: Performance under different values of $N$**
> >
> > | $N$ | **AUROC** | **F1-score** | **IoU** | Accuracy | Recall | Precision |
> > |-----|-----------|--------------|---------|----------|--------|-----------|
> > | 2   | 87.8      | 54.4         | 37.3    | 89.4     | 57.6   | 51.5      |
> > | 4   | 88.2      | 54.8         | 37.7    | 89.5     | 58.1   | 51.8      |
> > | 6   | 87.8      | 53.9         | 36.9    | 88.1     | 63.7   | 46.8      |
> >
> > **Table 3: Performance under different values of $\alpha$**
> >
> > | $\alpha$ | **AUROC** | **F1-score** | **IoU** | Accuracy | Recall | Precision |
> > |----------|-----------|--------------|---------|----------|--------|-----------|
> > | 0.2      | 87.6      | 54.0         | 37.0    | 88.7     | 60.5   | 48.8      |
> > | 0.5      | 88.2      | 54.8         | 37.7    | 89.5     | 58.1   | 51.8      |
> > | 1        | 87.0      | 50.9         | 34.1    | 89.9     | 47.7   | 54.6      |
> > | 2        | 87.1      | 52.6         | 35.7    | 89.6     | 52.8   | 52.4      |
> >
> > **Table 4: Performance under different values of $\beta$**
> >
> > | $\beta$ | **AUROC** | **F1-score** | **IoU** | Accuracy | Recall | Precision |
> > |---------|-----------|--------------|---------|----------|--------|-----------|
> > | 2       | 87.3      | 53.3         | 36.3    | 90.0     | 52.3   | 54.3      |
> > | 0.2     | 87.7      | 54.5         | 37.5    | 90.0     | 55.0   | 54.1      |
> > | $2 \times 10^{-2}$ | 87.6 | 53.9         | 36.8    | 89.5     | 56.3   | 51.6      |
> > | $2 \times 10^{-4}$ | 88.2 | 54.8         | 37.7    | 89.5     | 58.1   | 51.8      |
> > | $2 \times 10^{-5}$ | 87.5 | 53.3         | 36.4    | 88.8     | 58.4   | 49.0      |
> >
> > [1] Liu J, et al. Heterogeneous risk minimization. ICML 2021.
> >
> > [2] Sinha A, et al. Certifying Some Distributional Robustness with Principled Adversarial Training. ICLR 2018.
> >
> > [3] Zhang H, et al. Theoretically principled trade-off between robustness and accuracy. ICML 2019.
> >
> > [4] Tsipras D, et al. Robustness May Be at Odds with Accuracy. ICLR 2019.

---

### Official Review · Reviewer_QCJb · 2025-11-08

**Soundness:** 3
**Presentation:** 3
**Contribution:** 2
**Rating:** 6
**Confidence:** 4

**Summary:**

This paper proposes Spatiotemporal Distributionally Robust Optimization (STDRO), an optimization framework to enhance cross-patient generalization in EEG-based seizure detection and classification. Its functionality involves constructing and learning uncertainty sets in Distributionally Robust Optimization (DRO) that explicitly incorporate the spatiotemporal structure of EEG signals. Extensive experiments on the TUSZ and CHB-MIT datasets demonstrate consistent improvements over state-of-the-art baselines.

**Strengths:**

As a optimization-centric approach, STDRO seamlessly integrates with pre-training (e.g., VQ-MTM) and different types of network architectures.

Experiments are comprehensive, covering multiple datasets and clip durations. Ablation studies validates the benefit of each component (spatiotemporal structure and stability).

STDRO's uncertainty sets are data-adaptive, leveraging EEG's spatial correlations and temporal continuity, which is well-motivated.

**Weaknesses:**

While the method is empirically strong, given it is an optimization based approach, I would encourage the authors to provide more theoretical analysis on the benefit of the proposed approach for robustness and generalization.

Currently the work lacks comparison with other DRO-based EEG decoding approaches such as [1], which raises doubt on the contribution of this work. It also lacks discussion with EEG decoding approaches that also target robustness improvement [2].

I would encourage the authors to explore the sensitivity of the proposed work towards some of the other hyper-parameters such as alpha, beta etc. Which is currently missing.

[1] Distributionally robust cross subject EEG decoding, ECAI 2023
[2] Replay with stochastic neural transformation for online continual eeg classification, BIBM 2023

**Questions:**

Please see above section.

---

> ### Author Response · Authors · 2025-11-26
> **Response to Reviewer QCJb**
>
> We sincerely appreciate your recognition of our work’s merit and your insightful suggestions for improvement. We carefully address your concerns as follows
>
> 1. Theoretical analysis.
>
> Thank you for your valuable suggestion. We have supplemented a theoretical analysis of the proposed spatiotemporal uncertainty set in Appendix F. The analysis shows that if the spatiotemporally structured uncertainty set captures the intrinsic low-dimensional manifold of EEG data, the distributionally robust generalization bound can be tightened compared to standard DRO with Wasserstein balls. The core idea is to leverage the manifold assumption of the data ($m$-dimensional over the $D$-dimensional space, $m\ll D$) to show that when our distance metric aligns with the manifold, the worst-case distribution lies near the manifold, and the generalization bound becomes tighter than standard DRO when the worst-case distributions are constrained on the manifold. The informal statement is shown below:
>
> **Theorem 1** (Generalization Bounds, informal). For any $0<\delta<1$, with probability at least $1 - \delta$ , we have
>
> $\sup_{Q\in\mathcal{H}\_{\text{ST}}} \mathbb{E}\_{(x,y)\sim Q}[\ell(\hat{\theta};x,y)] \le \sup\_{Q:W\_{c_w}(Q, P_n)\le\rho} \mathbb{E}\_{(x,y)\sim Q}[\ell(\hat{\theta};x,y)] + \mathfrak{R}\_n(\mathcal{F}_{\text{adv}}) + \sqrt{\frac{\ln(1/\delta)}{2n}},$
>
> where $\mathfrak{R}\_n(\mathcal{F}\_{\text{adv}})$ is the Rademacher complexity of the adversarial loss class.
> When the worst-case distributions are constrained on the manifold, we have $\mathfrak{R}\_n(\mathcal{F}_{\text{adv}}) = \mathcal{O}\Big(\sqrt{\frac{m}{n}} \Big)$,
> which is smaller than the standard DRO complexity bound $\mathcal{O}\left(\sqrt{\frac{D}{n}}\right)$ given $m\ll D$.
>
> Complete details can be found in Appendix F.
>
> 2. EEG decoding literature.
>
> Thank you for your valuable references. We have incorporated the discussion of these papers into the revised paper, as quoted below:
>
> *"In EEG decoding, there are also works exploring cross-subject problems under the perspective of online continula learning (Duan et al., 2023a)."*
>
> *"Duan et al. (2023b) explored DRO in EEG decoding tasks by introducing dynamically evolved data distributions via Wasserstein gradient flows, while their approach does not exploit the intrinsic spatiotemporal structure of EEG signals as our method."*
>
> For comparison with Duan et al. (2023b), we improve DRO for EEG data from a different perspective, i.e., the spatiotemporal structure of data for constructing uncertainty set, while they do not exploit the intrinsic structure of data. Additionally, we primarily focus on seizure analysis. We have added the comparative discussion in the revised paper. As they do not release the code and there is no shared task, we do not compare in experiments.
>
> 3. Hyperparameter sensitivity.
>
> Thank you for your suggestion. We have added the sensitivity analysis of parameter $\alpha$, $\beta$ and $N$ in Appendix E, and the results are quoted below.
>
> **Table 1: Performance under different values of $N$**
>
> | $N$ | **AUROC** | **F1-score** | **IoU** | Accuracy | Recall | Precision |
> |-----|-----------|--------------|---------|----------|--------|-----------|
> | 2   | 87.8      | 54.4         | 37.3    | 89.4     | 57.6   | 51.5      |
> | 4   | 88.2      | 54.8         | 37.7    | 89.5     | 58.1   | 51.8      |
> | 6   | 87.8      | 53.9         | 36.9    | 88.1     | 63.7   | 46.8      |
>
> **Table 2: Performance under different values of $\alpha$**
>
> | $\alpha$ | **AUROC** | **F1-score** | **IoU** | Accuracy | Recall | Precision |
> |----------|-----------|--------------|---------|----------|--------|----------|
> | 0.2      | 87.6      | 54.0         | 37.0    | 88.7     | 60.5   | 48.8      |
> | 0.5      | 88.2      | 54.8         | 37.7    | 89.5     | 58.1   | 51.8      |
> | 1        | 87.0      | 50.9         | 34.1    | 89.9     | 47.7   | 54.6      |
> | 2        | 87.1      | 52.6         | 35.7    | 89.6     | 52.8   | 52.4      |
>
> **Table 3: Performance under different values of $\beta$**
>
> | $\beta$ | **AUROC** | **F1-score** | **IoU** | Accuracy | Recall | Precision |
> |---------|-----------|--------------|---------|----------|--------|-----------|
> | 2       | 87.3      | 53.3         | 36.3    | 90.0     | 52.3   | 54.3      |
> | 0.2     | 87.7      | 54.5         | 37.5    | 90.0     | 55.0   | 54.1      |
> | $2 \times 10^{-2}$ | 87.6 | 53.9         | 36.8    | 89.5     | 56.3   | 51.6      |
> | $2 \times 10^{-4}$ | 88.2 | 54.8         | 37.7    | 89.5     | 58.1   | 51.8      |
> | $2 \times 10^{-5}$ | 87.5 | 53.3         | 36.4    | 88.8     | 58.4   | 49.0      |
>
> [1] Sinha A, et al. Certifying Some Distributional Robustness with Principled Adversarial Training. ICLR 2018.
>
> [2] Duan T, et al. Distributionally Robust Cross Subject EEG Decoding. ECAI 2023.
>
> [3] Duan T, et al. Replay with stochastic neural transformation for online continual eeg classification. BIBM 2023.

---

### Meta-Review · Area_Chair_wejo · 2026-01-06

**Summary:**

This paper proposes SpatioTemporal Distributionally Robust Optimization (STDRO), a plug-and-play distributionally robust optimization framework for improving cross-patient EEG seizure detection/classification. STDRO constructs a spatiotemporally structured uncertainty set and adds a group-wise stability objective that updates covariate weights to reduce performance gaps across patient groups. The approach is evaluated on seizure benchmarks and is positioned as complementary to architectures and pretraining.

The forum shows substantial enthusiasm from one reviewer (hGu2: 8) and a positive-but-marginal stance from another (QCJb: 6), both citing the idea as sensible and empirically supported. However, there are two high-confidence negative reviews (fEhV: 2, conf 5; n1TW: 0, conf 5) that raise serious concerns about (i) clarity/coherence and objective specification, (ii) optimization complexity and tuning burden relative to gains, and (iii) protocol/statistical validity (splits, significance, variability). While the authors responded with additional theory, sensitivity analysis, added metrics, a new task (DOD-H sleep staging), and clarifications about splits and runtime, key doubts remain about whether the method is (a) sufficiently well-specified and interpretable to be reproducible and (b) clearly worth the complexity and runtime overhead in a way that is convincingly demonstrated and statistically grounded. Given the persistence and confidence of the negative concerns and the lack of reviewer score updates after rebuttal, the risk of accepting a method that is not clearly communicated and not robustly validated is too high.

**Reviewer Concerns:**

## Reviewer QCJb
### Addressed
- Need for theoretical analysis: Authors added an Appendix F argument (informal theorem) claiming a tighter DRO generalization bound under a manifold assumption when the spatiotemporal metric aligns with the data manifold.
- Hyperparameter sensitivity: Added sensitivity analysis for α,β,γ with AUROC/F1/IoU/Accuracy/Recall/Precision.
- Related work discussion: Added discussion of prior DRO EEG decoding and other robustness work.
### Still outstanding / partially addressed
- Lack of direct empirical comparison to prior DRO EEG decoding: Authors did not run experiments vs. Duan et al. (ECAI 2023) due to missing code/shared tasks. While reasonable, the absence leaves residual uncertainty about incremental value relative to closest DRO-for-EEG approaches.

---

## Reviewer hGu2
### Addressed
- “Under the hood” analyses: Added confusion matrix (worst class GN improved), and group-wise seizure characteristic visualization.
- Generality beyond seizure: Added an additional EEG task with improvements when applied to GraphS4former.
- Complexity/cost: Provided runtime estimates  and argued overhead is small compared to pretraining costs; graph constructed in advance; epoch budgeting comparable.

### Still outstanding / partially addressed
- Interpretability of the learned uncertainty set: Authors give high-level DRO intuition, but the practical meaning of the learned uncertainty set remains largely qualitative.
- Precision/recall tradeoff explanation: Response attributes it to random variation. This is plausible but not fully satisfying given robustness objectives can systematically shift operating points.

---

## Reviewer fEhV
`This is the most substantive and decisive critique.`
### Addressed
- Splitting protocol concerns: Authors clarified they follow prior works and used TUSZ v1.5.2 without official train/val split; CHB-MIT patient-based split consistent with prior description.
- Multiple runs / statistical robustness: Authors claim to have updated ablations with multiple seeds and added IoU in sensitivity analysis. They also mention multiple runs for the new DOD-H experiment.
- Cost: Provided explicit runtime overhead and argued it is acceptable.
### Still outstanding / not convincingly resolved
- Objective specification and interpretability (core method clarity): fEhV argued the objective is underspecified, stability term/patient grouping arbitrary, and M_t unclear. Authors provided a mathematical description, but the explanation still does not fully resolve why this particular parameterization and stability update is principled, or how it should be tuned and diagnosed in practice.
- Complexity vs gains: Even accepting the overhead argument, the reviewer’s critique is about added bi-level complexity for moderate/variable gains, with missing insight into convergence/tuning stability. The response asserts training stability similar to baselines but provides limited quantitative evidence .
- Variability and comparability: Authors state variability is expected and seen in prior work, but this does not substitute for demonstrating consistent effect sizes with confidence intervals across the main benchmark settings, especially given the reviewer’s concern that some gains might be within noise.

---

## Reviewer n1TW
### Addressed (partially)
- Authors clarified terminology and explained “spatial” as electrode/channel geometry and non-Euclidean structure, with citations.
### Still outstanding
- Presentation/coherence: n1TW’s critique is primarily about the manuscript being confusing and overloaded. The rebuttal explains that the terms are standard, but does not clearly show that the paper’s exposition has been substantially simplified/rewritten. Without an explicit reviewer follow-up, it is hard to conclude this concern would be resolved for readers.

**Reviewer Scores:**

`Reviewer QCJb`
- Likely change: 6 → 6
- Their main asks (theory, sensitivity, citations) were addressed. A full discussion could push them slightly upward if convinced about protocol/statistics, but lack of direct DRO comparison may keep them at 6.

---

`Reviewer hGu2`
- Likely change: 8 → 8
- Authors added exactly the analyses requested and expanded to another EEG task; this reviewer would likely remain strongly positive.

---

`Reviewer fEhV`
- Likely change: 2 → 4
- The authors did respond materially on splits, multiple runs (at least for ablations), cost, and added IoU. With discussion, the score might rise modestly from 2 to 4 but the core concerns about method clarity, tuning burden, and whether gains are robust across settings likely remain for this high-confidence reviewer.

---

`Reviewer n1TW`
- Likely change: 0 → 2
- Terminology confusions are addressed, which could move them off a “strong reject.” However, their central complaint is readability/coherence; unless substantial rewriting is demonstrated and acknowledged, discussion alone likely cannot raise this into an accept.

---

Even under optimistic discussion adjustments, we would still have two strong positives but also at least one (and likely two) below-threshold reviewers with high confidence. Given the weight of clarity/reproducibility and statistical rigor concerns for an optimization-heavy method, I do not support acceptance.

---

### Decision · Program_Chairs · 2026-01-26

Reject